# Origin of Circumpolar Deep Water intruding onto the Amundsen and Bellingshausen Sea continental shelves

Yoshihiro Nakayama[1], Dimitris Menemenlis [1], Hong Zhang[1,2], Michael Schodlok[1,2] & Eric Rignot[1,3]

Melting of West Antarctic ice shelves is enhanced by Circumpolar Deep Water (CDW) intruding onto the Amundsen and Bellingshausen Seas (ABS) continental shelves. Despite existing studies of cross-shelf and on-shelf CDW transports, CDW pathways onto the ABS originating from further offshore have never been investigated. Here, we investigate CDW pathways onto the ABS using a regional ocean model. Simulated CDW tracers from a zonal section across 67°S (S04P) circulate along the Antarctic Circumpolar Current (ACC) and Ross Gyre (RG) and travel into ABS continental shelf after 3–5 years, but source locations are shifted westward by ~900 km along S04P in 2001–2006 compared to 2009–2014. We find that simulated on- and off-shelf CDW is ~0.1–0.2 °C warmer in the 2009–2014 case than in the 2001–2006 case together with changes in simulated ocean circulation. These differences are primarily caused by lateral, rather than surface, boundary conditions, implying that large-scale atmospheric and ocean circulations are able to control CDW pathways and thus off- and on-shelf CDW properties.

[1] Jet Propulsion Laboratory, California Institute of Technology, 4800 Oak Grove Drive, Pasadena, CA 91109, USA. [2] Joint Institute for Regional Earth System Science and Engineering, University of Los Angeles, 4242 Young HallBox 957228Los Angeles, CA 90095, USA. [3] Department of Earth System Science, Croul Hall, University of California, Irvine, Irvine, CA 92697, USA. Correspondence and requests for materials should be addressed to Y.N. (email: Yoshihiro.Nakayama@jpl.nasa.gov)

Generally, Circumpolar Deep Water (CDW) circulates along the Antarctic Circumpolar Current (ACC) and Ross Gyre[1], some of which is carried along the southernmost branch of ACC and travels into the Amundsen Sea (AS) and Bellingshausen Sea (BS) (white arrows in Fig. 1). The CDW, intruding onto the AS and BS continental shelves via submarine glacial troughs (see, e.g., refs. [2,3]), causes rapid melting and thinning of the ice shelves of the West Antarctic Ice Sheet[4-6], contributing significantly to ongoing sea level rise through an increased discharge of grounded ice[7]. Despite existing works on cross-shelf and on-shelf CDW transports (see, e.g., refs. [8-14]), there have not been any studies investigating CDW pathways onto the AS and BS originating from further offshore.

Here, we investigate CDW pathways by releasing passive tracers representing CDW from World Ocean Circulation Experiment (WOCE) S04P section located at ~67°S. The WOCE S04P section was occupied in 1992 and 2011[15] and was measured again in 2018 under the GOSHIP (Global Ocean Ship-based Hydrographic Investigations Program). We use a regional AS and BS configuration of the Massachusetts Institute of Technology general circulation model (MITgcm) similar to ref. [16] but with a nominal horizontal grid spacing of ~1/12° (~2.5 km on the AS continental shelf) and some adjustments to model parameters (Supplementary Table 1). Similar to ref. [16], this model is capable of simulating along-slope undercurrent[13,14] and on-shelf CDW flow through major submarine glacial troughs located at the continental shelf break. Atmospheric forcing and lateral boundary conditions are provided by ongoing ECCO (Estimating the Circulation and Climate of the Ocean) LLC270 optimization (see ref. [16] and Model configuration for details). We use six virtual passive tracers (TR1 to TR6) to investigate CDW pathways onto the AS and BS continental shelves from different locations. For all tracers, initial tracer concentrations are set to 100.0 for water in the purple or orange boxes (located near the northern or eastern boundaries in Fig. 1) along 67°S or 135°W with potential density of 27.60–27.80 (Fig. 2) and no tracer restoring is applied. We conduct a 6-year model simulation from 2009 to 2014, hereinafter CTRL (2009–2014). Sensitivity experiments on CDW pathways to other years of atmospheric and lateral boundary forcings are also conducted (Supplementary Table 2). We find that simulated

CDW tracer pathways and off- and on-shelf CDW properties are different in 2009–2014 compared to 2001–2006. Our results demonstrate that large-scale atmospheric and ocean circulations are able to control CDW pathways and thus off- and on-shelf CDW properties.

## Results

**Spreading CDW tracer from WOCE S04P section (67°S).** The CTRL (2009–2014) case reproduces many features of ocean circulation, sea ice distribution, and ice shelf basal melt rates, in good agreement with observations (see Model evaluation for details). For our domain, the CTRL (2009–2014) Ross Gyre strength (Fig. 3), defined as the maximum stream function near the western model boundary, is 21 Sv (1 Sv = $10^6$ m$^3$ s$^{-1}$), consistent with previous studies (see, e.g., refs. [17,18]). For the simulated vertical sections along 67°S (Fig. 2), we are able to reproduce a temperature maximum located at ~500 m with warm upper CDW (~2 °C) at the eastern part of the section (75°−130°W) and a deepening and shallowing of isotherms and isohalines between, respectively, 140°–90°W and 90°–70°W. We note that the simulated slope of the 27.8 isopycnal in the eastern part is gentler than that in observations (Fig. 2), which is caused by a similar bias at the northern model boundary (Supplementary Figure 1). Simulated temperature and salinity at 552 m depth on the AS and BS continental shelves in 2010 are warmer and fresher than observations in 2007, 2009, and 2010; and simulated intruding CDW properties, for example, in the eastern AS and central BS are warmer by ~0.2 °C and ~0.3 °C, respectively (see refs. [3,19,20] and Supplementary Figures 2a, b).

CDW tracers (TR1, TR2, TR3, and TR4) released along 67°S flow southward along the ACC and are slowly advected southwestward (Fig. 4, Supplementary Movie 1) mostly following isopycnals. It takes 3–5 years for these CDW tracers to reach the AS and BS and flow into ice shelf cavities. For all the tracers on the AS continental shelf after 5 years of model simulation (January 2014), TR3 has the maximum contribution (59%) and TR1, TR2, and TR4 contribute 3, 6, and 30%, respectively. For all the tracers on the BS continental shelf after 5 years of model simulation (January 2014), the TR3 contribution is once again

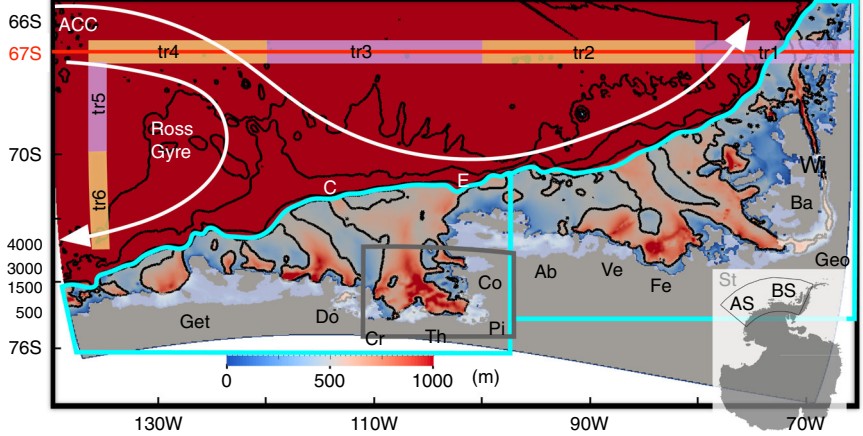

**Fig. 1** Model bathymetry (color) and initial release locations of tracers representing Circumpolar Deep Water (CDW) (orange and purple boxes). Bathymetric contours of 500, 1500, 3000, and 4000 m are shown with black contours. The inset (right bottom) shows Antarctica with the region surrounded by black line denoting the model domain. The red line represents the vertical section shown in Fig. 2. White arrows indicate the Ross Gyre and the southern extent of the Antarctic Circumpolar Current (ACC)[1]. CDW tracers are integrated in the Amundsen Sea (AS) and Bellingshausen Sea (BS) domains enclosed by cyan lines indicating the continental shelf regions south of 700 m isobaths along continental shelf break. The time series of area-averaged CDW temperature and heat content for the eastern AS region are calculated for the region enclosed by the gray line. AS and BS denote the Amundsen Sea and Bellingshausen Sea. Letters E and C denote the submarine glacial troughs located on the eastern and central AS continental shelf, respectively. Locations of ice shelves are shown with white patches and acronyms are summarized in Supplementary Table 3

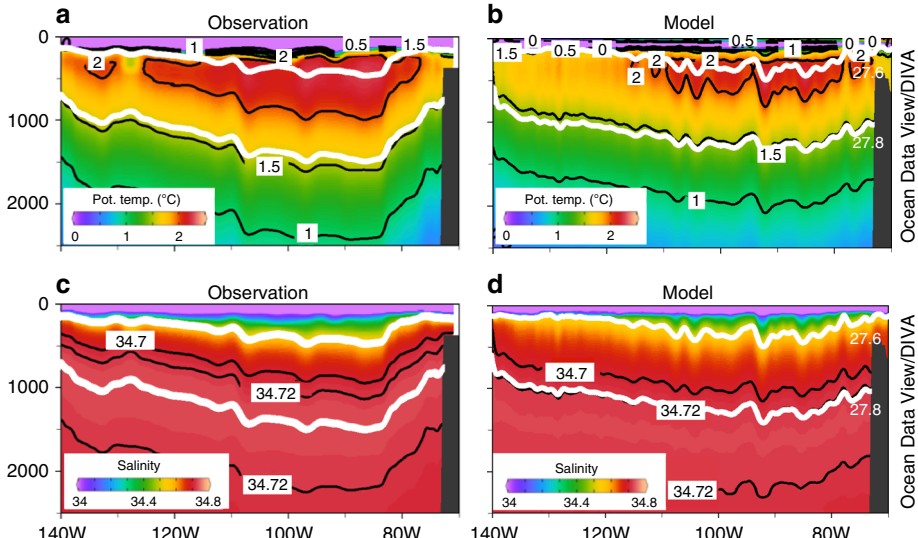

**Fig. 2** Observed and simulated sections of potential temperature and salinity in Jan 2011 along World Ocean Circulation Experiment (WOCE) S04P section (67°S). For all panels, potential temperature or salinity contours are shown with black lines with potential density contours of 27.60 and 27.80 (white)

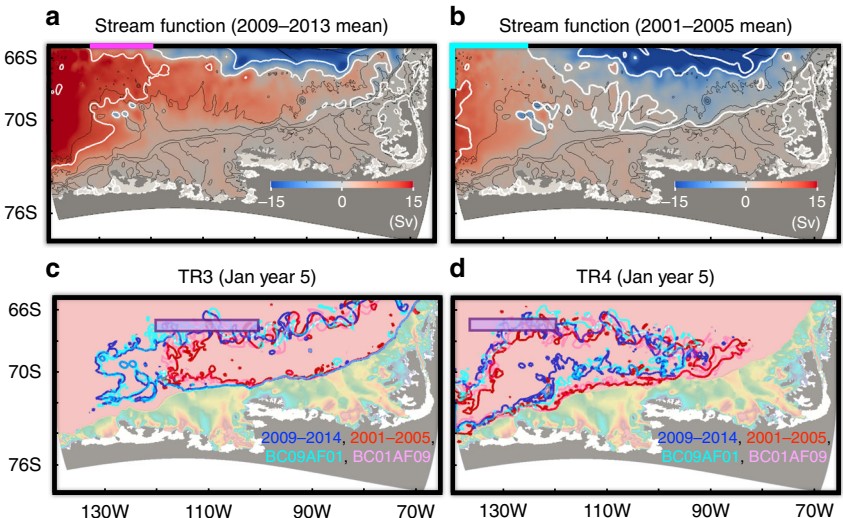

**Fig. 3** The 5-year mean stream functions and contours of vertically integrated tracer concentration of 2000 for TR3 and TR4 after 5 years of model simulation. The 5-year mean stream functions for **a** 2009–2013 and **b** 2001–2005 with counters of 20, 10, 0, −10, and −20 Sv are shown with bathymetric contours of 500, 1500, 3000, and 4000 m (black contours). Contours of vertically integrated tracer concentration of 2000 for **c** TR3 and **d** TR4 after 5 years of model simulation for the CTRL (2009–2014), 2001–2006, BC09AF01, and BC01AF09 cases are shown. For **c**, **d** bathymetry is shown with light colors in the same way as Fig. 1

largest (38%) and TR1, TR2, and TR4 contribute 23, 30, and 7%, respectively. TR5, and TR6 behave differently as the watermass is trapped in the Ross Gyre (2010) and is mostly advected out of our model domain through the western boundary (Fig. 4m–r).

Sequential snapshots of vertically integrated TR3 concentration describe a more detailed view of CDW transport from further offshore onto the AS continental shelf. In the first 5–7 months (Fig. 5a, b), CDW tracer is advected southward in the region with water depth deeper than ~4000 m. As CDW approaches the continental shelf break (after 9–15 months, Fig. 5c–f), a bathymetric ridge (red arrow in the Fig. 5a) assists CDW transport towards the AS continental shelf (Fig. 5d, e). This ridge seems to be important for separating the strong core of Ross Gyre circulation (with stream function larger than 10 Sv) from the weaker circulation extending to ~70°W (Fig. 3a) in the CTRL (2009–2014) case. As the TR3 reaches the continental shelf break with water depths of 500–1500 m (17–23 months), TR3 is transported eastward by an undercurrent (Supplementary Figure 3 and magenta arrows in Fig. 5g, h) as shown by previous studies[13,14] and intrudes onto the AS continental shelf via submarine glacial troughs.

To test the sensitivity of CDW pathways to other years of atmospheric and/or lateral boundary forcings, we perform three sensitivity experiments (Supplementary Table 2). The 2001–2006 case uses atmospheric forcing and lateral boundary conditions from 2001 to 2006. To separate the influence of atmospheric and lateral boundary forcings, we additionally perform the BC01AF09 and BC09AF01 cases where we use atmospheric forcing from 2009 to 2014 and 2001 to 2006 and lateral boundary conditions from 2001 to 2006 and 2009 to 2014, respectively.

The 2001–2006 case shows stronger eastward transport of CDW tracer along the ACC than the CTRL(2009–2014) case (e.g., contours of vertically integrated tracer concentration of 2000 in Fig. 3c, d) and CDW intruding onto the AS and BS continental

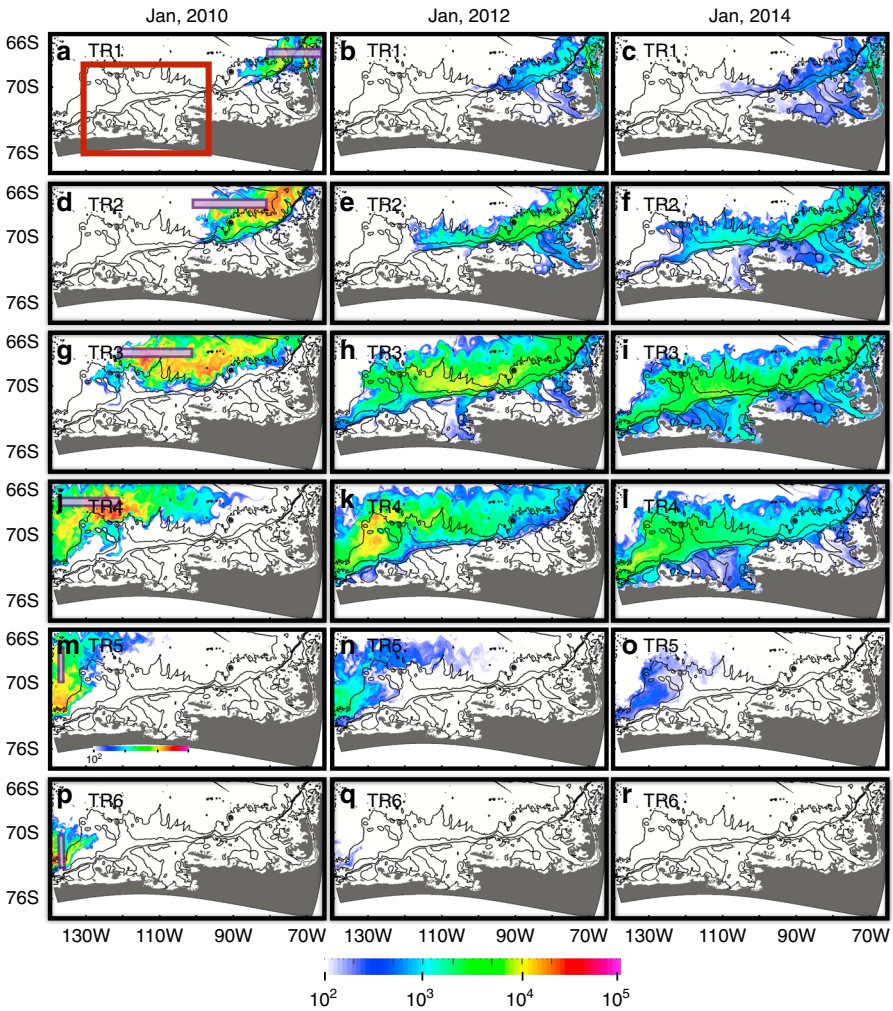

**Fig. 4** Monthly mean spatial distributions of vertically integrated tracers (TR) representing CDW after 1 (Jan 2010), 3 (Jan 2012), and 5 (Jan 2014) years of model simulation. **a–c** TR1, **d–f** TR2, **g–i** TR3, **j–l** TR4, **m–o** TR5, and **p–r** TR6 are used to investigate the CDW pathways from different locations and initial locations of these tracers are shown by the purple boxes in the left panels. Close up of the region enclosed by the red rectangle (**a**) is shown in Fig. 5. For all panels, bathymetric contours of 500, 1500, 3000, and 4000 m are shown with thin black contours

shelves originates from further west of the 67°S section (Figs. 3c, d, 4 and Supplementary Figure 4). For all tracers on the AS continental shelf after 5 years of model simulation (January 2006), the maximum contribution (54%) of tracers on the AS continental shelf is from TR4, as opposed to the CTRL (2009–2014) case described above where the maximum contribution originates from TR3. Sections TR1, TR2, TR3, TR5, and TR6 contribute 4, 1, 10, 20, and 11%, respectively, for the 2001–2006 case. For all the tracers on the BS continental shelf after 5 years of model simulation (January 2006), TR3 has the maximum amount (37%), similar to the CTRL (2009–2014) case, and TR1, TR2, TR4, TR5, and TR6 are 23, 12, 26, 1, and 1%, respectively. We also conduct experiments with restoring tracers (see Supplementary Figures 5 and 6 and Additional tracer experiments) showing that spreading pathways of CDW tracers remain similar throughout the model integration for all sensitivity experiments.

Here, we show that different lateral boundary conditions are the main reason for the different CDW pathways, while the influence of model domain atmospheric forcing on CDW tracer pathways is small based on atmospheric forcing versus lateral boundary condition experiments. CDW tracer distributions (e.g., TR3 and TR4) of the CTRL (2009–2014) case are similar to those in the BC09AF01 case, while they are clearly different from those in the BC01AF09 case whose distributions are similar to the

2001–2006 case (Figs. 3c, d, 4, and Supplementary Figure 4). These differences are caused by different circulation patterns (e.g., Ross Gyre strength and southern extent of ACC). The simulated Ross Gyre transports are 21 Sv and 13 Sv, respectively, for the CTRL (2009–2014) and 2001–2006 cases. A stronger Ross Gyre is also observed in the ECCO LLC270 optimization, which provides lateral boundary conditions for the regional simulations (see Model evaluation in Supplementary).

**Reason for off-shelf CDW warming**. Along with these differences in CDW tracer pathways, off-shelf CDW properties show warming of ~0.2 °C off the AS and BS continental shelves in the CTRL (2009–2014) case compared to the 2001–2006 case (Fig. 6 and Supplementary Figure 7). The off-shelf CDW potential temperature of the 2001–2006 case is similar to that in the BC01AF09 case, while off-shelf CDW potential temperature of the 2001–2006 case is clearly different from the BC09AF01 and 2009–2014 cases (Fig. 6). This means that lateral boundary condition changes dictate changes in large-scale ocean circulation and thus water source for the regions off the AS and BS. Similar to the behavior of spreading CDW tracers (Fig. 4), CDW warming off the AS continental shelf develops gradually over time. The warmest CDW is simulated during the last 2 years of

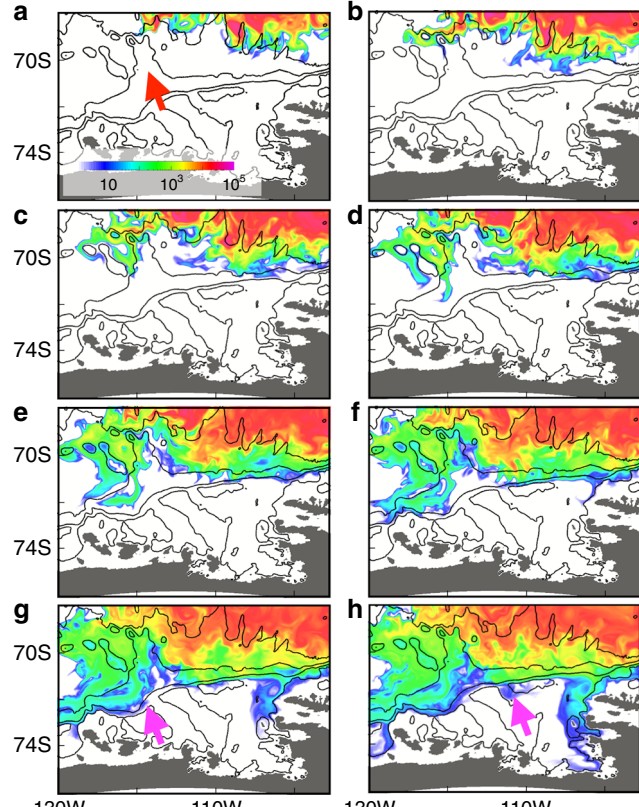

**Fig. 5** Monthly mean spatial distributions of vertically integrated TR3. Horizontal distributions of vertical integrated tracer after **a** 5, **b** 7, **c** 9, **d** 11, **e** 13, **f** 15, **g** 17, and **h** 19 months of simulation for the CTRL(2009–2014) case are shown. Bathymetric contours of 500, 1500, 3000, and 4000 m are shown with thin black contours. The red arrow indicates bathymetric ridge and magenta arrows indicate eastward transport of TR3 by the undercurrent

the CTRL (2009–2014) experiment (Supplementary Figure 8). For the region off the BS, off-shelf CDW stays warm after 1 year of model integration (Supplementary Figure 8).

This off-shelf CDW warming can be explained by the strengthened Ross Gyre circulation inside the regional domain and warmer temperature at the northern boundary. The strengthened Ross Gyre circulation allows CDW to approach the AS and BS continental shelves. By following iso-barotropic stream function contours, CDW off the AS is originated from ~120° to 130°W of the northern boundary for CTRL (2009–2014), while CDW off the AS is originated from ~125° to 140°W of the northern boundary and 65.5° to 68°S of the western boundary for the 2001–2006 case (highlighted by pink and cyan lines in Fig. 3a, b). The shift of the CDW source further to the east means warmer CDW southwards transport because CDW properties at the northern model boundary become generally warmer eastward (Fig. 2). In addition, most of the northern model boundary is warmer by ~0.1–0.2 °C in the CTRL (2009–2014) case than in the 2001–2006 case, indicating additional warming of CDW source (Fig. 6 and Supplementary Figure 7) likely caused by the slight southern shift of ACC (Supplementary Figure 9).

**Impact of off-shelf warming on on-shelf CDW properties**. The simulated warm off-shelf CDW penetrates onto the eastern AS and BS continental shelves showing on-shelf warming of 0.1–0.2 °C especially along the pathway of CDW on-shelf intrusion (Fig. 6). We compare volume-averaged CDW (defined to be the watermass with salinity higher than 34.65 (e.g., Supplementary Figures 2 and 8)) potential temperature on the eastern AS continental shelf in the vicinity of Pine Island Glacier (in the region enclosed by gray lines in Fig. 1). The CTRL (2009–2014) and BC09AF01 cases both with 2009–2014 lateral boundary forcing show warmer on-shelf CDW properties than those in 2001–2006 and BC01AF09 cases (Figs. 6 and 7). Averaged CDW potential temperature differences for the last 2 years of model simulations for this region caused by different atmospheric forcing are 0.04 °C (CTRL (2009–2014)–BC09AF01) and 0.004 °C (BC01AF09 and 2001–2006). In contrast, averaged

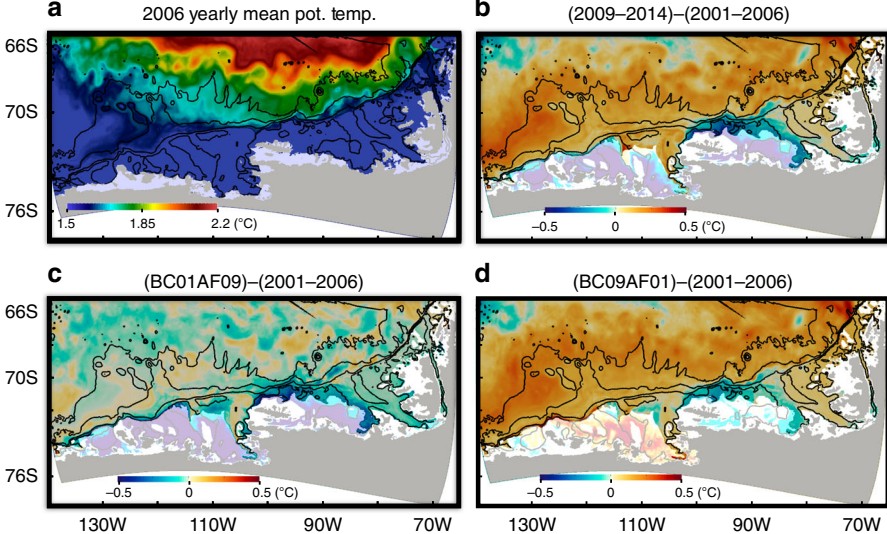

**Fig. 6** Yearly mean 409 m potential temperature in 2006 and mean potential temperature differences between CTRL (2009–2014) and 2001–2006 cases, BC01AF09 and 2001–2006 cases, and BC09AF01 and 2001–2006 cases (negative indicates colder water in 2001–2006 case). For **a**, yearly mean 409 m potential temperature in 2006 is shown and for **b-d**, annual mean differences are calculated using the results from year 6. Regions with CTRL (2009–2014) potential temperature lower than 1 °C are shaded with transparent white because potential temperature difference can be strongly affected by slight changes of thermocline depths in these regions

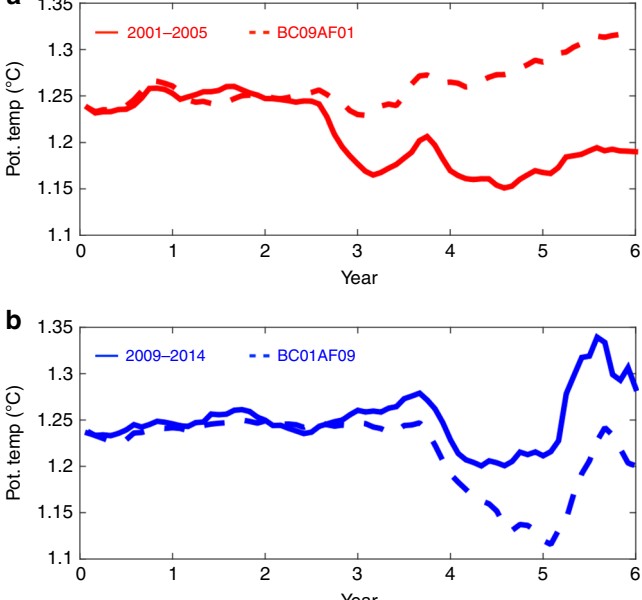

**Fig. 7** Time series of volume-averaged CDW (with salinity higher than 34.65) potential temperature for the eastern AS region regions (enclosed by the gray lines in Fig. 1) near Pine Island Glacier. Time series are shown for **a** 2001–2006 (red) and BC09AF01 (red dashed) cases and **b** CTRL (2009–2014) (blue) and BC01AF09 (blue dashed) cases. The X-axes show years from the start of simulations

CDW potential temperature differences for the last 2 years of model simulations caused by different lateral boundary forcing for this region are 0.08 °C (CTRL (2009–2014)−BC01AF09) and 0.12 °C (BC09AF01−2001–2006) (Fig. 7). This means that the lateral boundary forcing strongly influences on-shelf CDW potential temperature.

We note that on-shelf CDW can change its volume and its property. Recent studies (see, e.g., refs. [3,25]) focused on volume changes and showed that CDW volume reduction (or thickening of winter water (WW) by ~100–150 m) led to less CDW inflow into Pine Island Glacier cavity and reduction of the melt rate by ~50% in 2012. Consistent with these observations, the simulated vertically integrated heat content for the eastern AS region decreases between 2012 and 2014 (Supplementary Figure 10). Vertically integrated heat content is dominantly controlled by the depth of thermocline and thus atmospheric forcing with little impact of CDW temperature. For example, a heat content increase due to 0.1 °C CDW warming is equivalent to deepening of thermocline by 10 m, which is much smaller than observed changes, assuming 300 m thick CDW and temperature difference of 3 °C between WW and CDW. This study, in contrast, focuses on CDW property changes. Although vertically integrated heat content is shown to control ice shelf melt rates (see, e.g., refs. [3,25]), CDW temperature at grounding lines, for example, may also be important for controlling glacial retreats and thus future evolutions of glaciers.

**Observed CDW warming between 2012 and 2014.** Here, we compare our model results with hydrographic measurements in the AS available in 2009[21], 2010[19], 2011[22], 2012[3,23], and 2014[24]. We show that both data and model show existence of warmer off-shelf and on-shelf CDW properties between 2012 and 2014 compared to 2009–2011.

For the region off the eastern trough, off-shelf (defined as water depth deeper than 1000 m) CDW properties observed between 2012 and 2014 are 0.1–0.2 °C warmer than those observed between 2009 and 2011 for the entire CDW layer (350 to 600 m depth, Supplementary Figures 11d–e). On-shelf CDW properties are highly variable horizontally and vertically[13,19] but showing generally warmer on-shelf CDW properties in 2012–2014 than those in 2009–2011, with an average warming of 0.12 °C in the depth range of 450–600 m (Supplementary Figure 11e). For the region on and off the central trough, some CDW properties in 2012–2014 are warmer than those in 2009–2011, despite the limited number of observations available in this area (Supplementary Figures. 11b, c). To compare model results with observations, we sample the model in the same manner as the observations, obtaining vertical profiles of simulated potential temperature at the nearest locations and closest times to the observed profiles (Supplementary Figures 11f–i). The simulated results are generally similar (Supplementary Figures 11f–i) in terms of CDW properties, showing 0.1 °C and 0.05 °C CDW warming in 2012–2014 for the off- and on-shelf regions of the eastern trough (Supplementary Figures 11h–i), respectively, and existence of warmer CDW properties in 2012–2014 than those in 2009–2011, despite the large variability for the off- and on-shelf regions of the central trough (Supplementary Figures 11f–g). These features are caused by simulated off-shelf CDW warming (Fig. 6 and Supplementary Figure 7). In addition, mooring records located in the central trough and at the Pine Island Glacier front also show CDW warming between 2012 and 2013 and after January 2013, respectively [25].

We note that simulated WW in these regions is thinner than observations by 100-200 m, which may provide easier on-shelf access for off-shelf CDW. However, the fact that both data and model show warming of off- and on-shelf CDW between 2012 and 2014 on the eastern AS implies that our proposed mechanism, that large-scale ocean circulation controls off-shelf CDW properties leading to on-shelf CDW warming, seems realistic.

For the region off the BS, temperature in the Marguerite Trough is repeatedly observed under Antarctica LTER (Long Term Ecological Research) program and CDW temperature is colder than average for years 2001, 2003, 2004, and 2005, while CDW temperature is warmer than average for years 2002, 2009, 2010, 2011, and 2012, consistent with our model results except for 2002 [26].

## Discussion

With good agreement between model and observations representing off- and on-shelf CDW warming (Supplementary Figure 11), our simulations may present realistic off-shelf CDW transport towards the AS and BS continental shelves. However, in our regional simulations, off-shelf CDW warming is primarily controlled by large-scale ocean circulation and changes in mid-latitude CDW properties, which is controlled by the difference in lateral boundary conditions. Here, we analyze ECCO LLC270 global optimization and investigate the difference between CTRL (2009–2014) and 2001–2006 lateral boundary forcings.

The ECCO LLC270 global optimization shows a relatively stable Ross Gyre strength (defined as the maximum stream function in the Ross Sea) between 2001 and 2006 and 2009 and 2014 with mean Ross Gyre strengths of 28 Sv and 35 Sv, respectively (Supplementary Figure 12a). CDW temperature (409 m potential temperature) at the location of the regional model northern boundary (65.5°S) also shows warming with an average temperature of 1.9 °C and 2.2 °C for 2001–2006 and 2009–2014, respectively (Supplementary Figure 13). Considering that time series of Ross Gyre strength and CDW temperature at the northern model boundary show strengthening and warming,

respectively, with similar temporal patterns (Supplementary Figure 12 and 13) and the Ross Gyre extends more to the east in 2009–2014 than in 2001–2006 with stronger southwestward transport of warm CDW (Supplementary Figure 9), it seems likely that the Ross Gyre circulation and CDW temperature at the regional model northern boundary are dynamically linked.

The fact that off-shelf CDW warming is primarily controlled by lateral boundary forcing and thus large-scale ocean circulation and changes in mid-latitude CDW properties also means that these changes are caused by large-scale wind forcing and/or watermass formation. We analyze the relations with El Niño–Southern Oscillation (ENSO) and Southern Annular Mode (SAM) variabilities, since these variabilities have large impacts on the atmospheric circulation over the Southern Ocean. Based on the time series of SAM and ENSO indices (see Supplementary Figure 12b and Model configuration for details), the time-mean SAM index is 0.21 and 0.48 for the 2001–2006 and 2009–2014 periods, respectively, while the time-mean ENSO index is 0.19 and -0.22 for 2001–2006 and 2009–2014, respectively. Therefore, the 2001–2006 period corresponds to weaker SAM and El Niño years, while the 2009–2014 period corresponds to stronger SAM and La Niña years. However, no significant correlations can be found between Ross Gyre strength and SAM or ENSO indices. A recent study[27] using sea level anomalies of the Southern Ocean show that Ross Gyre strength is strongly influenced by local wind curl and local wind over the Ross Sea is significantly correlated with the SAM index implying a connection between Ross Gyre strength and SAM. We note that a strong connection between cross-shelf CDW intrusions and the large-scale atmosphere has also been previously suggested[3,28]. These two studies describe an atmospheric mechanism whereby a stationary Rossby wave, forced by tropical sea surface temperature, directly modulates wind stress over the AS continental shelf break and induces a localized ocean response. This mechanism is fundamentally different from our finding in that we show the importance of large-scale ocean circulation instead of localized ocean response for determining off- and on-shelf CDW properties.

We also note that many other on-shelf processes modify CDW properties (see, e.g., refs. [3,11,26,29–32]), some of which are possibly related to SAM and ENSO in different ways. For example, previous studies show strong correlations between La Niña/positive SAM events and on-shelf CDW warming (e.g., 2008–2014) on the BS continental shelf[26,29–32], while CDW intrusion onto the AS continental shelf is weakened during the 2012–2013 La Niña event, with a decline of Pine Island Glacier melt rate by ~50%[3,25].

In this study, we show that CDW tracers released from the WOCE S04P section (67°S) circulate along the ACC and Ross Gyre and travel into the AS and BS after 3–5 years. We demonstrate that large-scale ocean circulation and changes in mid-latitude CDW properties are able to change CDW pathways and control off-shelf CDW characteristics by ~0.2 °C and this off-shelf CDW feeds onto the AS and BS and determines on-shelf CDW properties, proposing a mechanism which determines on-shelf CDW properties on the AS and BS shelves. Consistent with existing observations, this off-shelf CDW penetrates onto the AS and BS continental shelves and leads to on-shelf warming between 2012 and 2014, at least for the region close to the continental shelf breaks. Multidecadal warming of Antarctic water is also observed[33] showing a similar spatial pattern to simulated off-shelf warming. The ongoing trend to a more positive SAM[34,35] shifts the westerlies southward, which may modulate large-scale ocean circulation and Ross Gyre strength. Such off-shelf changes may lead to off-shelf and intruding CDW warming, which possibly enhances the melting of West Antarctic glaciers and impacts their ongoing retreat (see, e.g. ref.[36]).

## Methods

**Model configuration.** We use a regional AS and BS configuration of the MITgcm, which includes dynamic/thermodynamic sea ice[37] and thermodynamic ice shelf[38] capabilities based on refs.[39,40]. The model domain contains the AS and BS (Fig. 1). Horizontal grid is extracted from a global LLC1080 configuration of the MITgcm. LLC1080 has the nominal horizontal grid spacing of 1/12°, compared to 1° for the LLC90 grid used for the ECCO version 4 solution (see ref.[41] for details). The LLC1080 grid uses a latitude–longitude grid north of 70°S and a bipolar grid south of 70°S (see ref.[41] for details). Horizontal grid spacing in the AS and BS domain is 2–3 km. The vertical discretization comprises 50 levels varying in thickness from 10 m near the surface, 70–90 m in the 500–1000 m depth range, and 450 m at the deepest level of 6000 m. Lateral boundary conditions are provided by ongoing ECCO LLC270 optimization. LLC270 is the grid selected for the next-generation ECCO ocean state estimation and has the nominal horizontal grid spacing of 1/3°, compared to 1° for the LLC90 grid used for the ECCO version 4 solution[41].

Model bathymetry is based on International Bathymetric Chart of the Southern Ocean (IBCSO[42]) with recent updates of more accurate bathymetry for the region near Crosson and Dotson ice shelves[43]. Model ice draft is based on Antarctic Bedrock Mapping (BEDMAP-2[44]). In addition, as shown by refs.[16,45], several icebergs are grounded off the Bear Peninsula along the 400 m isobath forming a barrier of grounded icebergs and landfast ice limiting sea ice transport between the eastern and central AS. Thus, we treat the region as a barrier in the sea ice model similar to ref.[16]. The model domain includes the George VI, Wilkins, Bach, Stange, Ferrigno, Venable, Abbot, Cosgrove, Pine Island, Thwaites, Crosson, Dotson, and Getz ice shelves (Fig. 1).

Model parameters used for this study are similar to the optimized simulation of ref.[16] but some model parameters for atmospheric forcing and sea ice are adjusted to more typically used values (Table S1). Initial conditions are derived from a 10-year (2001–2010) spin-up, integrated from rest and from January World Ocean Atlas 2009 temperature[46] and salinity[47] fields. Similar to ref.[16], atmospheric forcing is provided by ongoing ECCO LLC270 optimization, which is based on ERA-Interim[48] and has been adjusted using the ECCO adjoint-model-based methodology[49]. Boundary conditions are also provided by ECCO LLC270 optimization. There is no additional freshwater run off, that is, all calving icebergs are assumed to be transported and melt outside the regional AS and BS domains.

The ENSO historical Oceanic Niño Index is acquired from the NOAA (National Oceanic and Atmospheric Administration) Center for Weather and Climate Prediction, which is based on the 3-month running mean of ERSST.v4 SST anomalies in the Niño 3.4 region. The Marshall SAM station-based index is obtained as monthly values from NCAR (National Center for Atmospheric Research), which is available at reliable quality from 1979 to present.

**Model evaluation.** The ocean circulation of ECCO LLC270 global optimization, which is used as the lateral boundary conditions, shows the Ross Gyre of ~35 Sv and 28 Sv for 2009–2014 mean and 2001–2006 mean (Supplementary Figures 9 and 12), respectively, consistent with previous studies (see, e.g., refs.[17,18]). In these cases, Ross Gyre strength is defined as the maximum stream function in the Ross Gyre and gyre strengths are larger than those shown in this study, because the center of the gyre is located outside (further to the west) of our model domain.

For sea ice, the simulated mean winter (September) sea ice extent is similar to satellite-based estimates[50], while mean summer (March) sea ice extent is underestimated by 83% between 2009 and 2013. For the static ice shelf component, we adjust turbulent heat and salt exchange coefficients for individual ice shelves to constrain the mean melt rates to match with the satellite-based estimates of basal melt rates[5] similar to ref.[16] (see Supplementary Table 3). Mean basal melt rates for George VI, Pine Island, Thwaites, and Getz Ice Shelves are 88, 98, 100, and 117 Gt yr⁻¹ for 2009–2013, respectively, consistent with other satellite-based estimates; for example, see refs.[4,5] (Supplementary Table 3).

**Additional tracer experiments.** All tracers used in this study are set to 100.0 initially for water in the purple or orange boxes (located near the northern or eastern boundaries in Fig. 1) along 67°S or 135°W with potential density of 27.60–27.80 (Fig. 2) and no tracer restoring is imposed. Tracers are only released initially in order to better represent the time scale required for these tracers to reach the continental shelves and regions of where these tracers spread (e.g., Fig. 4). Here, we conduct 6 additional tracer-release experiments, identical to the initial release experiments, but where tracers are restored to a concentration of 1.0 at every time step with a restoring time scale of 1 h in the purple and orange boxes (Fig. 1) along 67°S or 135°W with potential density of 27.60–27.80. Horizontal distributions of average tracer concentration between 27.6 and 27.8 isopycnals (Supplementary Figures 5 and 6) show qualitatively similar patterns compared to initially released CDW tracers (Fig. 4 and Supplementary Figure 5). This indicates that the spreading pathways of CDW tracers remain similar throughout each of these sensitivity experiments. After 5 years of integration, TR3 concentration on the AS continental shelf in the vicinity of the eastern trough and on the north-eastern BS continental shelf is ~40%, indicating that nearly half of the intruding CDW in these locations originates from the WOCE S04P section (67°S).

**Data availability**. The model code, some processing tools, and instructions for reproducing these results are available at http://wwwcvs.mitgcm.org/viewvc/MITgcm/MITgcm_contrib/antarctic/llc_1080_ABS.

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

## Acknowledgements

The research was carried out at the Jet Propulsion Laboratory, California Institute of Technology, under a contract with the National Aeronautics and Space Administration (NASA) and at the University of California, Irvine. Support was provided by an appointment to the NASA Postdoctoral Program; the NASA Cryosphere program; and

the NASA Modeling, Analysis, and Prediction program. Computations were carried out at the NASA Advanced Supercomputing facilities. We thank Shigeru Aoki, Karen M. Assmann, and Catherine C. Walker for their useful comments and suggestions. Some of the figures are created with software Paraview[51] and Ocean Data View [52].

## Author contributions

Y.N. conceived the study, conducted the ocean modeling, and wrote the initial draft of the paper. D.M. and H.Z. contributed to the global LLC270 optimization and helped with regional model set-up. Y.N., D.M., H.Z., M.S., and E.R. contributed to interpreting the results, discussion of the associated dynamics, and refinement of the paper.

## Additional information

**Competing interests:** The authors declare no competing interests.

