## [Peer Review file · Nature Communications]

Reviewers' comments:

Reviewer #1 (Remarks to the Author):

This manuscript studies the origin of the Circumpolar Deep Water that intrudes the Amundsen and Bellingshausen Seas in West Antarctica and ultimately melts the ice sheet. The main tool used is a regional simulation, where sensitivities to boundary conditions and atmospheric forcing are tested. The potentially 'paradigm shift' element of this work, eventually warranting publication in Nature, is that the authors suggest that what really matters for continental shelf seas heat interannual variability is the variability of offshore water properties, not the variability of the fluxes at the continental shelf break separating shelf seas from the Southern Ocean, or local continental shelf seas forcing.

Despite the use of elaborate and complex tools, I find the manuscript to be very confusing and surprisingly lacking in diagnostics and quantifications supporting the main conclusions.

Using tracer releases, the authors do show that the boundary conditions of their regional configuration matter over long timescales in setting a background of open ocean properties which will ultimately feed continental shelves. On this point, comparison of these model results with observations are weak or not clearly shown in the present manuscript, but let's assume such a link does exist.

How efficient is the system in transposing this offshore variability onto the continental shelves? To me the authors analysis on this point is not clear nor convincing. Despite having model outputs from which budget analysis can be computed, the authors rely on presenting timeseries at the continental shelf edge, not on the seas themselves, or using tracer experiments that do not appear to quantify the impact for shelf seas.

I think these interesting simulations warrant further analysis and comparisons with observations, and I encourage the authors to persevere in their investigations before re-submitting the manuscript.

Below are more detailed comments/requests for clarifications.

Variability within the chosen 5 years time window:

Are the atmospheric forcing and boundary conditions relatively constant over the 5 years time windows you have chosen? If not, aren't you not running into aliasing issues?

Tracer experiments:

Are tracers released at the start of the 5 years simulations? Is the release continual over the 5 years period? In the former case, how does one dissociate initial value problems from genuine semi-decadal variability in tracer sources, e.g. do you obtain the same patterns if you initiate tracer release in year 2 or 3 of the scenarios?

For each continental basin, what is the ultimate % volume occupied by the released tracer? In other words, should we care about a 0.2 degree decrease in ocean temperature if it only concerns 10% of all inflowing waters on the continental shelf?

Line 17: WOCE SO4P section is likely to be mostly unknown to most readers, making it a very weak reference point for the abstract, and 67S seems to imply to the uneducated reader circum-Antarctic, which it isn't. You may want to reduce it to a more generic version, e.g. 'a zonal section across the Southern Pacific' or simplify by saying that all water entering the ABS system originates from the South western Pacific sector of the Southern Ocean?

Line 19: Observations indicate warm conditions from 2006-2011, colder conditions before, and a cold anomaly after [Dutrieux et al., 2014; Jenkins et al., 2016; Webber et al., 2017]□, so the timing seems off. You do mention this later in the text...

Line 23-26: It would be more convincing if you explain what line of evidence leads you to advance

this hypothesis. Something like: "Past work indicated that Here, ... analysis and ... diagnostics show that, instead, far field ocean properties and circulation control more than half of the off-shelf CDW warming".

Line 25: I find the wording misleading. Surely you do not mean to imply that on the timescales of a few years, the CDW properties themselves are evolving due to the atmospheric forcing? What you mean to say, perhaps, is that atmospheric forcing brings it closer or further away from the continental shelf?

Line 27: the "proposed mechanism" is not clearly explained at this stage.

Line 34: "white arrows"?

Figure 2 and line 61: It's not obvious that the model is doing such a great job. Why is the modeled temperature so far off from observations close to the northern boundary? Aren't you initializing with essentially WOCE data? Even the geostrophic circulation seems way off (much stronger density gradients in the Eastern part in the observations). Any comments on that? Would you get a better fit by looking at this section further north in the simulation? Or is the ACC path/Ross gyre just not that well represented in the model? This seems to be an important detail given your conclusions, so it would be nice to get a more detailed comparison, if that is possible. Figures S1a-b actually do not show a much better agreement between observed and measured temperatures, especially on the continental shelf. In a way it's ok to have model biases, but they need to be acknowledged.

Figure 5 has 8 panels, but only 7 timestamps. What is panel h showing? Month 19? Also, why looking at vertically integrated tracers? Is the tracer diffusion depth independent??

Line 79: Ok. So tracers are being advected. That is expected. Perhaps more interesting is the way they are advected. Is it really uniform? And in what sense? Is it uniform horizontally? Does it vary in the vertical? One would expect surface advection to be very different (at least in terms of magnitude) from that at 4000m depth. Isn't it?

In fact I am confused: the tracer is one of CDW (released within 2 isopycnals), not one at all depth in the model, correct? If so, why should we concern ourselves with 4000m depth horizontal tracer diffusion? Unless you mean advection is uniform in the deep parts of the basin?

Perhaps more completely, shouldn't we be looking at tracer modification and relevance for the overall heat/salt budget on the Amundsen and Bellingshausen basins?

Line 82: About the importance of the ridge: it may well play a strong role, but it is interesting to note that the streamfunction shown in figure 3a-b, for various forcing, shows very different tracer behaviors, with one scenario where tracer distribution do not seem dictated that much by the bathymetry. Do we still see the impact of the ridge in the 2001-2005 forcing experiment?

Line 85: the undercurrent shown in previous studies [Assmann et al., 2013; Walker et al., 2007] is located at the continental shelf break, not on the shelf as one arrow suggests. Velocity sections would be more revealing and persuading on that matter.

Line 115: this statement is really region dependent: perhaps choose a precise location and point to it.

Line 114-126: I am not convinced by the discrepancy between circulation and temperature boundary conditions: aren't they dynamically linked, i.e. ones comes with the other. If so it does not really make sense to discuss the difference between one and the other, unless you can run (tricky) additional experiments showing that one matters more than the other, etc.

Line 127: Figure S4 is really not sufficient to compare model and observations. Surely, given the

scale of the change you are claiming happens, there are more than 3 observed data points that can be looked at for comparison? Also, isn't the offshore signal different from the onshore one, at least in the Amundsen, where 2012-2014 appear to be colder than 2009-2011 [Webber et al., 2017]? What does that mean for your conclusions?

Line 131: Comparison with Marguerite Trough data is circumstantial at best: you basically fit a trend with 2 points (2 simulations), and you are not showing the results. That is fine, but not very conclusive.

Line 136: 'with good agreement with observations'. I really think you should refrain from such a statement. One reason is that it can be disputed. Another reason is that you do not need such agreement to investigate model physics and sensitivity.

Line 137: Again, isn't the 'warming of the northern boundary' dynamically linked to the strength of the Ross gyre? If so, is it significant that it is warming? Or is it more significant that the gyre is growing and weakening?

Line 153: Why concentrate on the eastern end of that section, where a slight shift in front position is going to lead to a large anomaly, but may not be a large scale signal. I think you can make a case that the change is large scale based on this comparison. That said, the issue here is that you are comparing 1992 to 2011, which is a different time and timescale than the one you are proposing here. Any hint 1992 represents your 2001-2005 era?

Figure S6 and line 157. Lots of questions here... If instead of using the last two years for comparisons, you used the last one, you would find a very different answer, especially in the 'Eastern trough', wouldn't you? Looking at these timeseries, I think it is fair to say that conditions at the shelf edge do vary based on both boundary conditions and atmospheric forcing. This is especially true in the "central trough", which appears to be much more variable, and where variability within one timeseries is at least as large as that over the entire 5 years.

But you do have a full model output, so why limit yourself to small boxes, and small boxes at the continental shelf edge at that? Why not looking at larger areas, and not at one depth (here 409m)?

I think you demonstrate reasonably well that the southern ocean part is sensitive to boundary conditions. But it is much less clear what the impact on the Amundsen and Bellingshausen heat/salt budgets are!

Of course offshore boundary conditions ought to matter, but ultimately off-shelf/on-shelf exchanges are also determining how much of the offshore variability makes its way onto the shelf.

Line 165-176: ?? Wait, so does your mechanism matter? Or does it not? Isn't it possible to quantify contributions from your model outputs?

Line 185-187: But aren't you finding that your 'offshore' mechanism is going against the 'shelf edge' mechanism? If so, shouldn't we expect a lessened impact of such SAM/ENSO links?

Assmann, K. M., A. Jenkins, D. R. Shoosmith, D. P. Walker, S. S. Jacobs, and K. W. Nicholls (2013), Variability of Circumpolar Deep Water transport onto the Amundsen Sea Continental shelf through a shelf break trough, *J. Geophys. Res. Ocean.*, 118(12), 6603–6620, doi:10.1002/2013JC008871. [online] Available from: <http://doi.wiley.com/10.1002/2013JC008871>

Dutrieux, P., J. De Rydt, A. Jenkins, P. R. Holland, H. K. Ha, S. H. Lee, E. J. Steig, Q. Ding, E. P.

Abrahamsen, and M. Schröder (2014), Strong Sensitivity of Pine Island Ice-Shelf Melting to Climatic Variability., *Science*, 3(7), 468–472, doi:10.1126/science.1244341. [online] Available from: <http://www.sciencemag.org/cgi/doi/10.1126/science.1244341> (Accessed 10 January 2014)

Jenkins, A., P. Dutrieux, S. Jacobs, E. Steig, H. Gudmundsson, J. Smith, and K. Heywood (2016), Decadal Ocean Forcing and Antarctic Ice Sheet Response: Lessons from the Amundsen Sea, *Oceanography*, 29(4), 106–117, doi:10.5670/oceanog.2016.103. [online] Available from: <http://www.tos.org/oceanography/article/decadal-ocean-forcing-and -antarctic-ice-sheet-response-lessons-from-the-amu>

Walker, D. P., M. A. Brandon, A. Jenkins, J. T. Allen, J. A. Dowdeswell, and J. Evans (2007), Oceanic heat transport onto the Amundsen Sea shelf through a submarine glacial trough, *Geophys. Res. Lett.*, 34(2), 1–4, doi:10.1029/2006GL028154. [online] Available from: <http://www.agu.org/pubs/crossref/2007/2006GL028154.shtml> (Accessed 13 March 2012)

Webber, B. G. M., K. J. Heywood, D. P. Stevens, P. Dutrieux, E. P. Abrahamsen, A. Jenkins, S. S. Jacobs, H. K. Ha, S. H. Lee, and T. W. Kim (2017), Mechanisms driving variability in the ocean forcing of Pine Island Glacier, *Nat. Commun.*, 8, 14507, doi:10.1038/ncomms14507. [online] Available from: <http://dx.doi.org/10.1038/ncomms14507>

Reviewer #2 (Remarks to the Author):

This paper uses a numerical ocean model to answer the question of where the CDW that floods the continental shelves of the Amundsen and Bellingshausen seas originates. The authors conclude that this water originates to the north, in a region around 110°W, 67°S, but that this source region has shifted eastward between 2001-2006 and 2009-2014. This shift in source region is associated with warming of CDW. The authors suggest that this shift in the CDW properties drives a significant portion of the melt rate of glaciers in West Antarctica, and that this may accelerate in future. These results are novel, and are certainly of interest to the oceanographic community focussed on this region. However, their relevance to a wider audience rests on whether the results are important for the melt rate of ice shelves in West Antarctica and thus sea level rise.

I find that this last point is not well supported by evidence presented in this paper, and somewhat at odds with existing literature. The authors ignore the question of how much CDW is driven onto the continental shelves and the associated thermocline depth, which is usually seen as more important than the temperature of the core CDW (e.g., Thoma et al., 2008; Jacobs et al., 2011, 2013; Dutrieux et al., 2014; Christianson et al., 2016; Webber et al., 2017). Jacobs et al. (2011) explicitly state that the small observed increase in CDW was most likely a secondary cause of the increase in melt rate observed, which was predominantly linked to the circulation strength. Meanwhile, the thermocline depth is often seen as crucial for determining the temperature of water able to reach the grounding line, depending on the bathymetry within the ice shelf cavity, as well as controlling the total heat available to melt ice within the cavity. This manuscript implies that changes in CDW temperature are dominant, or at least as important as dynamic changes generated at the shelf break and on the continental shelf; I am not convinced, and think that the remote changes in CDW temperature will play a relatively minor role in determining the melt rate of ice shelves around the Amundsen and Bellingshausen seas.

I am also perplexed by why the paper does not examine the CDW temperature and thermocline depth on the shelf, and how this changes between the two time periods considered. The argument is made that CDW has warmed due to a different origin, but whether this warming reaches the ice shelves is not clearly shown. Fig. S6 shows that the temperature at the shelf break increases steadily between 2009-2014. However, conditions on the continental shelf and especially close to Pine Island Glacier exhibited strong changes during this time period (Dutrieux et al., 2014; Christianson et al., 2016; Webber et al., 2017), largely dominated by cooling from 2009-2013 followed by a partial recovery. Therefore, either the model is failing to capture the variability on the Amundsen Sea continental shelf, or the conditions are not very sensitive to CDW temperature.

The authors allude to this in the paragraph beginning at line 165, but this paragraph does not really resolve the differences between this and previous studies: what role if any is there for winds within the domain to modify the flux of CDW and its properties within this model? What might explain the on-shelf cooling between 2012-2014 when the source CDW was warming? Does this model simulate this cooling or not, and if not, why not?

The analysis of what atmospheric changes cause the variability in CDW pathways and temperature is weak, which is unfortunate. It would be better to include more quantitative analysis of the difference in lateral boundary conditions between 2001-2006 and 2009-2014, and how SAM and ENSO might contribute. At the moment, the manuscript refers to the Walker & Gardner (2017) paper to state, rather imprecisely, that 2001-2006 corresponds to weaker SAM and El Niño years while 2009-2014 corresponds to stronger SAM and La Niña years. This argument would be greatly strengthened by timeseries of SAM and ENSO indices. In addition, it is not clear on what time scale the Ross Gyre might respond to such forcing, and whether it is SAM, ENSO, or their combination that drives the variability in Ross Gyre strength.

Line 127 & Fig. S4: What are the observations used for this comparison? There are quite a few other observational records in and around the Amundsen Sea. For example, Anna Wåhlin's group has maintained moorings in the "western trough" leading to Dotson & Getz ice shelves for many years (probably approaching a decade now). There have been multiple moored records in the Pine Island Trough region and of course lots of (summer) research cruises. Therefore, this analysis seems very limited in scope. There has also been analysis (Schmidtke et al., 2014, Science) of trends in temperature around Antarctica that should be cited in this paper.

Nature communication articles usually have subheadings – I think the manuscript would be improved by their inclusion.

References to "Supplementary for detail" are too vague – please direct the reader to specific locations in supplementary material.

Specific issues. There are a number of minor issues with phrasing that I have not listed here. I recommend a careful proof-read on resubmission. I have listed a few particular points below.

Line 13: "thinning rapidly... mainly caused by Circumpolar Deep Water..." – the ice shelves are melting rapidly due to CDW, but the thinning is due to a combination of melting and glacial dynamics, including the retrograde slope of the bedrock, which may be more important for the retreat in places.

Line 17 and elsewhere: I am not familiar with this WOCE section and I imagine that many readers will be even less so. Therefore, the use of this section needs some explanation and the longitudinal extent of the section (and, if different, the part used in this study) needs to be stated.

Line 34: typo: allows -> arrows

Line 68: I would say "are advected" not "advect"

Line 174: I believe that the mechanism described in the papers referenced relates to westerly winds at the shelf break, not their curl (which may also be important, but that is a different matter).

Reviewer #3 (Remarks to the Author):

This paper uses the results of a regional ocean-ice model to investigate the transport of Circumpolar Deep Water (CDW) from the ACC to the shelf break in the Amundsen and Bellingshausen seas. Once that water mass is on the continental shelf it fuels rapid melting of the floating ice shelves, thinning of which has led to acceleration of the outflow from the grounded ice sheet. Processes that transport CDW to the shelf edge have tended to be overlooked, with the inflow to the shelf and transport across it being the main focus of numerous earlier studies. The study reported here is therefore novel, the results are interesting, and there is certainly the potential to add important insight into how the ocean drives ice sheet change in West Antarctica. However, the authors do not deliver on that potential, opting for a simple "ocean warming, ice melting" story that does not really fit their results, rather than taking a more objective look at where and how this study can add to knowledge of what in reality is clearly a complex interaction of atmosphere-ice-ocean processes. I therefore cannot recommend publication of the manuscript in its current form.

The main problem is that the model results show warming of the CDW at the shelf edge over the 2009-2014 period versus the 2001-2006 period. Some observational support is given for this in the supplementary material, but the sources are not made clear, the figure is hard to decipher (see later comments) and it shows only three stations through the entire area. So while that may or may not be a valid result, a far better documented change is the cooling of the waters on the Amundsen Sea continental shelf that occurred during the same 2009-2014 period. That cooling, rather than any off-shelf warming, is what dictated the change in ice sheet melting that occurred at that time.

The on-shelf cooling is acknowledged in one sentence (lines 168-171), then completely glossed over, with the conclusions stating that the processes discussed in the paper that cause off-shelf warming "may play a dominant role in determining CDW properties intruding onto the continental shelves". This leads to a final (misleading) statement that since the difference between 2009-2014 and 2001-2006 was probably caused by a more positive SAM, and we expect SAM to be more positive in future, then we should expect warmer conditions and more rapid retreat of the ice sheet in future. But if the continental shelf was actually colder during the more positive SAM phase, when the off-shelf CDW was paradoxically warmer, isn't that more suggestive of other processes dominating the on-shelf conditions? If those cool conditions result from the positive SAM phase, wouldn't that suggest weaker forcing of the ice sheet in future when SAM is more positive?

Why not focus the paper on explaining the apparent paradox between off-shelf warming and on-shelf cooling in the Amundsen? That would make for a much more useful and insightful paper? A related result that is potentially really interesting, but that has been buried by the authors' apparent desire to say that everywhere is warming, is that the Amundsen and Bellingshausen seas may respond in anti-phase to SAM/ENSO variability. There are hints in the data that when the Bellingshausen warms the Amundsen cools and vice versa. If the model shows that, and the authors can explain it, then there is a result worth publishing. But the thrust of the paper suggests that the model doesn't actually capture the on-shelf cooling in 2009-2014 in the Amundsen, in which case such an investigation would not be possible.

Furthermore, I think there are a few issues with the experimental design, or at least the presentation of that design, that need clarification. The authors conclude that the main determinant of the differences in the model simulations are the boundary conditions they apply. That effectively means that the explanation for what caused the changes lies not in the model results they are reporting, but in the larger-scale model from which the boundary conditions were taken. Very little text is devoted to those other results, and no explanation for the change in circulation is offered. The closest we get to an explanation comes on lines 143-147, but that only says that strong SAM "may" strengthen the Ross Gyre, and seems likely to increase upwelling. However, that seems to come from other work. Why not look at the ECCO results and tell the reader what they show?

The other issue is the initial conditions, which are not mentioned at all in the main text. The methods section suggests that the initial conditions are the same for all experiments and are generated from a 10-year spin-up (with what boundary conditions?) of a model that starts from climatology. So what part of the 6-year runs reported in the paper is actually a transient response to a step change in boundary conditions? Are the CDW tracers continuously-released, or just introduced at the start? The figures suggest the latter. If that is the case, then the final distribution could be strongly dictated by the initial transient response, and that might be why the boundary conditions appear to exert the dominant control over their final distribution. Certainly Figure S6 suggests that the dominance of the boundary conditions over local surface forcing is not a robust or universal result. There are many times (in particular most of the time between year 1 and 4 on the time axis) when the difference between runs with different surface forcing is much larger than that between the runs with different boundary conditions.

There are also problems with a number of the figures that really need to be addressed:

Figure 1: What projection is used to display the main panel? The edges of the model domain appear to be both curvilinear and asymmetric, and to extend beyond the displayed box. So the area covered by the model domain is never really clear. The inset suggests the edges follow parallels and meridians, so why do they look so distorted on the main panel? The same comment applies to all the map-plane figures that follow.

Figure 3(c and d) and S3: It looks to me as if the inset boxes are incorrectly plotted on the main figure panels. That is misleading. The main panels indicate boxes that are on the continental shelf, but temperatures and isobaths (?) in the inset suggest that the box might actually be somewhere to the north-east of where it is indicated to be.

Figure S4: What is the source of these data? Why show so few stations from such a small region? If there really are only six stations, couldn't the comparison be shown in a more informative way? How about showing profiles? And what does the grey shading show? Presumably bathymetry, but what is the scale? Again, the form of the contours suggests that the box is not on the shelf.

Table S3 gives ranges in the last column, but didn't the cited paper give climatological mean values for each ice shelf? What do the ranges mean and where do they come from?

Response to the specific comments

*(Italic: comments from reviewer, * and bold :our reply)*

***We greatly appreciate all very helpful and insightful reviewers' comments.**

Reviewers' comments:

Reviewer #1 (Remarks to the Author):

This manuscript studies the origin of the Circumpolar Deep Water that intrudes the Amundsen and Bellingshausen Seas in West Antarctica and ultimately melts the ice sheet. The main tool used is a regional simulation, where sensitivities to boundary conditions and atmospheric forcing are tested. The potentially 'paradigm shift' element of this work, eventually warranting publication in Nature, is that the authors suggest that what really matters for continental shelf seas heat interannual variability is the variability of offshore water properties, not the variability of the fluxes at the continental shelf break separating shelf seas from the Southern Ocean, or local continental shelf seas forcing. Despite the use of elaborate and complex tools, I find the manuscript to be very confusing and surprisingly lacking in diagnostics and quantifications supporting the main conclusions.

Using tracer releases, the authors do show that the boundary conditions of their regional configuration matter over long timescales in setting a background of open ocean properties which will ultimately feed continental shelves. On this point, comparison of these model results with observations are weak or not clearly shown in the present manuscript, but let's assume such a link does exist.

*** In the revised manuscript, we conduct detailed model-data comparison (Lines 155-174 and Fig. S10).**

How efficient is the system in transposing this offshore variability onto the continental shelves? To me the authors analysis on this point is not clear nor convincing. Despite having model outputs from which budget analysis can be computed, the authors rely on presenting timeseries at the continental shelf edge, not on the seas themselves, or using tracer experiments that do not appear to quantify the impact for shelf seas. I think these interesting simulations warrant further analysis and comparisons with observations, and I encourage the authors to persevere in their investigations before re-submitting the manuscript.

*** In the revised manuscript, we modified 2 points. To get an idea of the efficiency of this system, we (1) show January mean bottom temperature difference between 2014 and 2006**

(Fig. 3e). This demonstrates that off-shelf warming indeed penetrates onto the continental shelves but the influence of warming is limited to the area close to the continental shelf break after 5 years of model simulation. We also show (2) detailed model-data comparison displaying that both on- and off-shelf CDW warming is observed supporting our model results (Lines 155-174 and Fig. S10).

However, we do not conduct budget analysis for quantifying the impact on shelf seas because this requires longer model integration (with and without off-shelf CDW warming) as we only see the on-shelf CDW warming in the limited area in the last 2 years of model simulations (Fig. 3e). In addition, we think that cross-shelf exchange is another difficult research question that cannot be easily addressed and requires different kind of model evaluations.

Below are more detailed comments/requests for clarifications.

Variability within the chosen 5 years time window: Are the atmospheric forcing and boundary conditions relatively constant over the 5 years time windows you have chosen? If not, aren't you not running into aliasing issues?

*** Off-shelf CDW temperature only shows a gradual change and off-shelf CDW warming can be seen clearly between 2012-2014 whenever it is sampled (e.g., Figs. S8). Thus, we are not running into aliasing issues.**

Tracer experiments:

Are tracers released at the start of the 5 years simulations? Is the release continual over the 5 years period? In the former case, how does one dissociate initial value problems from genuine semi-decadal variability in tracer sources, e.g. do you obtain the same patterns if you initiate tracer release in year 2 or 3 of the scenarios?

*** Tracers are released at the start of the 6-year simulations and no restoring is applied. In the revised manuscript, we also conduct 2 additional simulations with restoring tracers. These simulations show that we get similar patterns even with restoring tracers (Figs. S6 and S7). Manuscript is revised (Lines 450-465).**

For each continental basin, what is the ultimate % volume occupied by the released tracer? In other words, should we care about a 0.2 degree decrease in ocean temperature if it only concerns 10% of all inflowing waters on the continental shelf?

*** To answer your question, we conduct additional simulations with restoring tracers (Figs. S6 and S7). Ultimate % volume occupied by released (restoring) tracer is ~50% in the eastern trough in the AS and on the BS continental shelf (Lines 109-112 and 450-464). In terms of potential temperature, on-shelf warming of ~0.1-0.2°C is simulated.**

Line 17: WOCE S04P section is likely to be mostly unknown to most readers, making it a very weak reference point for the abstract, and 67S seems to imply to the uneducated reader circum-Antarctic, which it isn't. You may want to reduce it to a more generic version, e.g. 'a zonal section across the Southern Pacific' or simplify by saying that all water entering the ABS system originates from the South western Pacific sector of the Southern Ocean?

*** We replace “WOCE S04P section” with “a zonal section across the Southern Pacific”.**

Line 19: Observations indicate warm conditions from 2006-2011, colder conditions before, and a cold anomaly after [Dutrieux et al., 2014; Jenkins et al., 2016; Webber et al., 2017], so the timing seems off. You do mention this later in the text...

*** In the revised manuscript, we show that off-shelf CDW warms and on-shelf CDW (especially near Pine Island Glacier) cools during 2012-2014. We also conduct more detailed model-data comparison (See latter comments for detail).**

Line 23-26: It would be more convincing if you explain what line of evidence leads you to advance this hypothesis. Something like: “Past work indicated that Here, ... analysis and ... diagnostics show that, instead, far field ocean properties and circulation control more than half of the off-shelf CDW warming”.

*** We revise the manuscript as suggested (Lines 22-26).**

Line 25: I find the wording misleading. Surely you do not mean to imply that on the timescales of a few years, the CDW properties themselves are evolving due to the atmospheric forcing? What you mean to say, perhaps, is that atmospheric forcing brings it closer or further away from the continental shelf?

*** This line is removed from the manuscript.**

Line 27: the “proposed mechanism” is not clearly explained at this stage.

*** This part of the text is rephrased (Lines 26-28).**

Line 34: “white aRRows”?

*** We revise the manuscript as suggested.**

Figure 2 and line 61: It's not obvious that the model is doing such a great job. Why is the modeled temperature so far off from observations close to the northern boundary? Aren't you initializing with essentially WOCE data? Even the geostrophic circulation seems way off (much stronger density gradients in the Eastern part in the observations). Any comments on that? Would you get a better fit by looking at this section further north in the simulation? Or is the ACC path/Ross gyre just not that well represented in the model? This seems to be an important detail given your conclusions, so it would be nice to get a more detailed comparison, if that is possible.

*** Model is initialized with WOCE data but boundary conditions are based on ECCO LLC270 global optimization, which includes similar bias (Fig. S1). We explain this bias in the revised manuscript (Lines 66-68).**

Figures S1a-b actually do not show a much better agreement between observed and measured temperatures, especially on the continental shelf. In a way it's ok to have model biases, but they need to be acknowledged.

*** We revise the manuscript as suggested (Lines 68-71).**

Figure 5 has 8 panels, but only 7 timestamps. What is panel h showing? Month 19? Also, why looking at vertically integrated tracers? Is the tracer diffusion depth independent??

*** Panel h shows Month 19. We only release tracers from the initial condition since we would like to show (1) the time scale of how long it takes for these tracers to reach the continental shelves and (2) the region of influence where tracers spread (e.g., Fig. 3f). These features can be better represented with initially released tracers. For these cases, spatial distributions of tracers are rather more important than their concentrations and we show vertically integrated tracer contents. Tracer diffusion is depth independent. Manuscript has been revised (Lines 72-74 and Table S1).**

Line 79: Ok. So tracers are being advected. That is expected. Perhaps more interesting is the way they are advected. Is it really uniform? And in what sense? Is it uniform horizontally? Does it vary in the vertical? One would expect surface advection to be very different (at least in terms of magnitude) from that at 4000m depth. Isn't it? In fact I am confused: the tracer is one of CDW (released within 2 isopycnals), not one at all depth in the model, correct? If so, why should we concern ourselves with 4000m depth horizontal tracer diffusion? Unless you mean advection is uniform in the deep parts of the basin? Perhaps more completely, shouldn't we be looking at tracer modification and relevance for the overall heat/salt budget on the Amundsen and Bellingshausen basins?

*** We remove this part of the text from the revised manuscript. We only state in the revised manuscript that tracers “are slowly advected southwestward mostly following isopycnals”.**

We believe that confusing part is “the region deeper than ~4000 m”. We mean that CDW tracer is advected southward at the region with water depth deeper than ~4000 m. Tracer mostly advects along the isopycnal (Figs. S6 and S7). We revised the manuscript (Line 84).

Line 82: About the importance of the ridge: it may well play a strong role, but it is interesting to note that the streamfunction shown in figure 3a-b, for various forcing, shows very different tracer behaviors, with one scenario where tracer distribution do not seem dictated that much by the bathymetry. Do we still see the impact of the ridge in the 2001-2005 forcing experiment?

*** In both CTRL(2009-2014) and 2001-2006 cases, tracers are more advected southward along the bathymetric ridge (Figs. 5d-e, Figs.S2g, and i). However, as pointed out, we do not see the impact on the stream function in the 2001-2005 experiments. We clarified this point in the revised manuscript.**

Line 85: the undercurrent shown in previous studies [Assmann et al., 2013; Walker et al., 2007] is located at the continental shelf break, not on the shelf as one arrow suggests. Velocity sections would be more revealing and persuading on that matter.

*** We include horizontal section of velocity (Fig. S3, similar to Figs. 9 e and f in Assmann et al., 2013) in the revised manuscript.**

Line 115: this statement is really region dependent: perhaps choose a precise location and point to it.

*** We add Fig. 3e showing the difference between Jan 2014 and Jan2006 monthly mean potential temperature at 409 m.**

Line 114-126: I am not convinced by the discrepancy between circulation and temperature boundary conditions: aren't they dynamically linked, i.e. ones comes with the other. If so it does not really make sense to discuss the difference between one and the other, unless you can run (tricky) additional experiments showing that one matters more than the other, etc.

*** Circulation and temperature at the boundary are from ECCO LLC270 global optimization and these should be dynamically linked. We do not intend to separate these impact in the manuscript. We include further discussion in the revised manuscript (Lines 187-196)**

Line 127: Figure S4 is really not sufficient to compare model and observations. Surely, given the scale of the change you are claiming happens, there are more than 3 observed data points that can

looked at for comparison? Also, isn't the offshore signal different from the onshore one, at least in the Amundsen, where 2012-2014 appear to be colder than 2009-2011 [Webber et al., 2017]? What does that mean for your conclusions?

*** We include detailed model-data comparison (Fig. S10 and Lines 156-174). We also include discussion on the simulated near-continent cooling (Fig. 3e and Lines 140-154).**

Line 131: Comparison with Marguerite Trough data is circumstantial at best: you basically fit a trend with 2 points (2 simulations), and you are not showing the results. That is fine, but not very conclusive.

*** Fig. S4 in the original manuscript is replaced with Fig. S10. We include detailed model data comparison (Lines 156-174).**

Line 136: 'with good agreement with observations'. I really think you should refrain from such statement. One reason is that it can be disputed. Another reason is that you do not need such agreement to investigate model physics and sensitivity.

*** In the revised manuscript we rephrased the sentence and clarify what we reproduce well in the model (Lines 181-183).**

Line 137: Again, isn't the 'warming of the northern boundary' dynamically linked to the strength of the Ross gyre? If so, is it significant that it is warming? Or is it more significant that the gyre is growing and weakening?

*** We agree with the reviewer that warming of the northern boundary is dynamically linked with the Ross Gyre. We add additional discussion on this point (Lines 187-196). However, since these are dynamically linked, we could not easily separate these contributions and could not argue which is more important.**

Line 153: Why concentrate on the eastern end of that section, where a slight shift in front position is going to lead to a large anomaly, but may not be a large scale signal. I think you can make a case the the change is large scale based on this comparison. That said, the issue here is that you are comparing 1992 to 2011, which is a different time and timescale than the one you are proposing here. Any hint 1992 represents your 2001-2005 era?

*** We think this part is speculative and is removed.**

Figure S6 and line 157. Lots of questions here... If instead of using the last two years for comparisons, you used the last one, you would find a very different answer, especially in the 'Eastern trough', wouldn't you? Looking at these timeseries, I think it is fair to say that conditions at

the shelf edge do vary based on both boundary conditions and atmospheric forcing. This is especially true in the “central trough”, which appears to be much more variable, and where variability within one timeseries is at least as large as that over the entire 5 years. But you do have a full model output, so why limit yourself to small boxes, and small boxes at the continental shelf edge at that? Why not looking at larger areas, and not at one depth (here 409m)? I think you demonstrate reasonably well that the southern ocean part is sensitive to boundary conditions. But it is much less clear what the impact on the Amundsen and Bellingshausen heat/salt budgets are! Of course offshore boundary conditions ought to matter, but ultimately off-shelf/on-shelf exchanges are also determining how much of the offshore variability makes its way onto the shelf.

*** We agree with the reviewer. As suggested, we rephrased the abstract (Lines 22-25), main text (Lines 140-154), and conclusion (Lines 235-237).**

The main point of this paper is that (1) there is an off-shelf mechanism that modifies off-shelf CDW properties, (2) both in model and observations, a warming signal is observed both on and off the continental shelves, and (3) such off-shelf mechanism may play an important role in determining the on-shelf CDW properties. We think that these points are supported by our modifications in the revised manuscript.

We do not conduct heat and salt budget analysis crossing the shelf edge for this work, because we think that understanding the impact of off-shelf warming onto the continental shelf requires long-enough simulations with and without off-shelf warming. In addition, cross-shelf exchange is another difficult research question that cannot be easily addressed and requires different kind of model evaluations with special focus on the shelf break regions.

Line 165-176: ?? Wait, so does your mechanism matter? Or does it not? Isn't it possible to quantify contributions from your model outputs?

*** In this paper, we focus on the off-shelf mechanism that controls the off-shelf CDW properties, which penetrates onto the continental shelves (based on both data and model). It means that the proposed mechanism matters for on-shelf CDW properties as shown both in model and data. However, we are not able to quantify the contribution for ice shelf melting from our model output. We clarified this point in the revised manuscript (Lines 218-225)**

Line 185-187: But aren't you finding that your 'offshore' mechanism is going against the 'shelf edge' mechanism? If so, shouldn't we expect a lessened impact of such SAM/ENSO links?

*** In this paper, we focus on the off-shelf mechanism that controls off-shelf CDW properties,**

which penetrates onto the continental shelves (based on both data and model). We include further discussion on this point (Lines 218-225).

Jenkins, A., P. Dutrieux, S. Jacobs, E. Steig, H. Gudmundsson, J. Smith, and K. Heywood (2016), Decadal Ocean Forcing and Antarctic Ice Sheet Response: Lessons from the Amundsen Sea, *Oceanography*, 29(4), 106–117, doi:10.5670/oceanog.2016.103. [online] Available from: <http://www.tos.org/oceanography/article/decadal-ocean-forcing-and-antarctic-ice-sheet-response-lessons-from-the-amu>

Walker, D. P., M. A. Brandon, A. Jenkins, J. T. Allen, J. A. Dowdeswell, and J. Evans (2007), Oceanic heat transport onto the Amundsen Sea shelf through a submarine glacial trough, *Geophys. Res. Lett.*, 34(2), 1–4, doi:10.1029/2006GL028154. [online] Available from: <http://www.agu.org/pubs/crossref/2007/2006GL028154.shtml> (Accessed 13 March 2012)

Reviewer #2 (Remarks to the Author):

This paper uses a numerical ocean model to answer the question of where the CDW that floods the continental shelves of the Amundsen and Bellingshausen seas originates. The authors conclude that this water originates to the north, in a region around 110°W, 67°S, but that this source region has shifted eastward between 2001-2006 and 2009-2014. This shift in source region is associated with warming of CDW. The authors suggest that this shift in the CDW properties drives a significant portion of the melt rate of glaciers in West Antarctica, and that this may accelerate in future. These results are novel, and are certainly of interest to the oceanographic community focused on this region. However, their relevance to a wider audience rests on whether the results are important for the melt rate of ice shelves in West Antarctica and thus sea level rise.

I find that this last point is not well supported by evidence presented in this paper, and somewhat at odds with existing literature. The authors ignore the question of how much CDW is driven onto the continental shelves and the associated thermocline depth, which is usually seen as more important than the temperature of the core CDW (e.g., Thoma et al., 2008; Jacobs et al., 2011, 2013; Dutrieux et al., 2014; Christianson et al., 2016; Webber et al., 2017). Jacobs et al. (2011) explicitly state that the small observed increase in CDW was most likely a secondary cause of the increase in melt rate observed, which was predominantly linked to the circulation strength. Meanwhile, the thermocline depth is often seen as crucial for determining the temperature of water able to reach the grounding line, depending on the bathymetry within the ice shelf cavity, as well as controlling the total heat available to melt ice within the cavity. This manuscript implies that changes in CDW

temperature are dominant, or at least as important as dynamic changes generated at the shelf break and on the continental shelf; I am not convinced, and think that the remote changes in CDW temperature will play a relatively minor role in determining the melt rate of ice shelves around the Amundsen and Bellingshausen seas.

*** We agree with the reviewer that changes in source CDW property play a minor role in determining ice shelf melt rates on short time scales. However, given the long-term trend towards positive Southern Annular Mode, modified large-scale ocean circulation may exert a dominant control on the intruding CDW properties on longer time scales. But this is just a hypothesis, which is not addressed in this study. In the revised manuscript, we modified abstract and conclusion to clarify these points (Lines 25-28 and 238-241).**

I am also perplexed by why the paper does not examine the CDW temperature and thermocline depth on the shelf, and how this changes between the two time periods considered. The argument is made that CDW has warmed due to a different origin, but whether this warming reaches the ice shelves is not clearly shown. Fig. S6 shows that the temperature at the shelf break increases steadily between 2009-2014. However, conditions on the continental shelf and especially close to Pine Island Glacier exhibited strong changes during this time period (Dutrieux et al., 2014; Christianson et al., 2016; Webber et al., 2017), largely dominated by cooling from 2009-2013 followed by a partial recovery. Therefore, either the model is failing to capture the variability on the Amundsen Sea continental shelf, or the conditions are not very sensitive to CDW temperature. The authors allude to this in the paragraph beginning at line 165, but this paragraph does not really resolve the differences between this and previous studies: what role if any is there for winds within the domain to modify the flux of CDW and its properties within this model? What might explain the on-shelf cooling between 2012-2014 when the source CDW was warming? Does this model simulate this cooling or not, and if not, why not?

*** Since we focus on off-shelf pathway of CDW, we think that the question “what explains the on-shelf cooling between 2012-2014” remains for future work. However, we show the detailed model-data comparisons displaying off-shelf and on-shelf warming both present in the model and data (Fig. S10 and Lines 122-174). We show the difference between 2014 Jan and 2006 Jan 409-m potential temperature to present the impact of off-shelf CDW warming on on-shelf CDW properties (Fig. 3e and Lines 140-155) after 5 years of model integration. We also include further discussion related to previous studies on thickening of the Winter Water (Lines 151-154).**

The analysis of what atmospheric changes cause the variability in CDW pathways and temperature

is weak, which is unfortunate. It would be better to include more quantitative analysis of the difference in lateral boundary conditions between 2001-2006 and 2009-2014, and how SAM and ENSO might contribute. At the moment, the manuscript refers to the Walker & Gardner (2017) paper to state, rather imprecisely, that 2001-2006 corresponds to weaker SAM and El Niño years while 2009-2014 corresponds to stronger SAM and La Niña years. This argument would be greatly strengthened by timeseries of SAM and ENSO indices. In addition, it is not clear on what time scale the Ross Gyre might respond to such forcing, and whether it is SAM, ENSO, or their combination that drives the variability in Ross Gyre strength.

*** In response to this comment, we included one paragraph discussing the time series of ENSO, SAM, Ross Gyre strength, and temperature at the northern model boundary (Lines 181-196). We also add one citation (Armitage et al., accepted), which shows that (1) there are strong correlations between local wind curl and Ross Gyre strength and (2) local wind curl is significantly correlated with non-seasonal SAM index.**

Line 127 & Fig. S4: What are the observations used for this comparison? There are quite a few other observational records in and around the Amundsen Sea. For example, Anna Wåhlin's group has maintained moorings in the "western trough" leading to Dotson & Getz ice shelves for many years (probably approaching a decade now). There have been multiple moored records in the Pine Island Trough region and of course lots of (summer) research cruises. Therefore, this analysis seems very limited in scope. There has also been analysis (Schmidtko et al., 2014, Science) of trends in temperature around Antarctica that should be cited in this paper.

*** In the revised manuscript, we (1) clarified the source of the data (Lines 156-157), (2) include a new figure comparing observed and simulated vertical profiles focusing on the off-shelf and on-shelf CDW properties (Fig. S10), and (3) include Schmidtko et al., 2014 in Line 237. We do not include any comparisons for the western trough because we see little warming in that location in our simulation (Fig. 3e).**

Nature communication articles usually have subheadings – I think the manuscript would be improved by their inclusion.

*** We revised the manuscript as suggested.**

References to "Supplementary for detail" are too vague – please direct the reader to specific locations in supplementary material.

*** We revised the manuscript as suggested.**

Specific issues. There are a number of minor issues with phrasing that I have not listed here. I

recommend a careful proof-read on resubmission. I have listed a few particular points below.

Line 13: “thinning rapidly... mainly caused by Circumpolar Deep Water...” – the ice shelves are melting rapidly due to CDW, but the thinning is due to a combination of melting and glacial dynamics, including the retrograde slope of the bedrock, which may be more important for the retreat in places.

*** Thank you for pointing out. We replaced “thinning” with “melting”.**

Line 17 and elsewhere: I am not familiar with this WOCE section and I imagine that many readers will be even less so. Therefore, the use of this section needs some explanation and the longitudinal extent of the section (and, if different, the part used in this study) needs to be stated.

***Following reviewer 1, we replaced “WOCE S04P section” with “a zonal section across the Southern Pacific”**

Line 34: typo: allows -> arrows

*** We revised the manuscript as suggested.**

Line 68: I would say “are advected” not “advect”

*** We revised the manuscript as suggested.**

Line 174: I believe that the mechanism described in the papers referenced relates to westerly winds at the shelf break, not their curl (which may also be important, but that is a different matter).

*** As suggested, we replaced “wind stress curl” with “wind stress”.**

Reviewer #3 (Remarks to the Author):

This paper uses the results of a regional ocean-ice model to investigate the transport of Circumpolar Deep Water (CDW) from the ACC to the shelf break in the Amundsen and Bellingshausen seas. Once that water mass is on the continental shelf it fuels rapid melting of the floating ice shelves, thinning of which has led to acceleration of the outflow from the grounded ice sheet. Processes that transport CDW to the shelf edge have tended to be overlooked, with the inflow to the shelf and transport across it being the main focus of numerous earlier studies. The study reported here is therefore novel, the results are interesting, and there is certainly the potential to add important insight into how the ocean drives ice sheet change in West Antarctica. However, the authors do not deliver on that potential, opting for a simple “ocean warming, ice melting” story that does not really fit their results, rather than taking a more objective look at where and how this study can add to knowledge of what in reality is clearly a complex interaction of atmosphere-ice-ocean processes. I

therefore cannot recommend publication of the manuscript in its current form.

The main problem is that the model results show warming of the CDW at the shelf edge over the 2009-2014 period versus the 2001-2006 period. Some observational support is given for this in the supplementary material, but the sources are not made clear, the figure is hard to decipher (see later comments) and it shows only three stations through the entire area. So while that may or may not be a valid result, a far better documented change is the cooling of the waters on the Amundsen Sea continental shelf that occurred during the same 2009-2014 period. That cooling, rather than any off-shelf warming, is what dictated the change in ice sheet melting that occurred at that time.

*** In the revised manuscript, we (1) clarify the data source (Lines 156-158), (2) include a new figure comparing vertical profiles of available observations between 2009-2014 focusing on the off-shelf and on-shelf CDW properties (Fig. S10), and (3) include discussion on simulated warming on the continental shelf (Lines 123-139). Cooling of the AS continental shelf is simulated consistent with observations and we also clarify this point.**

The on-shelf cooling is acknowledged in one sentence (lines 168-171), then completely glossed over, with the conclusions stating that the processes discussed in the paper that cause off-shelf warming “may play a dominant role in determining CDW properties intruding onto the continental shelves”. This leads to a final (misleading) statement that since the difference between 2009-2014 and 2001-2006 was probably caused by a more positive SAM, and we expect SAM to be more positive in future, then we should expect warmer conditions and more rapid retreat of the ice sheet in future. But if the continental shelf was actually colder during the more positive SAM phase, when the off-shelf CDW was paradoxically warmer, isn't that more suggestive of other processes dominating the on-shelf conditions? If those cool conditions result from the positive SAM phase, wouldn't that suggest weaker forcing of the ice sheet in future when SAM is more positive?

*** In the revised manuscript, we (1) clarify that our model also simulates the colder near-continental shelf conditions during 2012-2014 (Lines 109-112, 151-154, Fig. 3e) and (2) show that both model and data show warming of off-shelf and on-shelf CDW properties (Lines 123-154, Fig. S10) at the same time.**

We also include additional discussion that on-shelf conditions likely respond differently to SAM and ENSO, possibly leading to on-shelf cooling (Lines 218-225).

Why not focus the paper on explaining the apparent paradox between off-shelf warming and on-shelf cooling in the Amundsen? That would make for a much more useful and insightful paper? A related result that is potentially really interesting, but that has been buried by the authors' apparent desire to

say that everywhere is warming, is that the Amundsen and Bellingshausen seas may respond in anti-phase to SAM/ENSO variability. There are hints in the data that when the Bellingshausen warms the Amundsen cools and vice versa. If the model shows that, and the authors can explain it, then there is a result worth publishing. But the thrust of the paper suggests that the model doesn't actually capture the on-shelf cooling in 2009-2014 in the Amundsen, in which case such an investigation would not be possible.

*** Thank you very much for your suggestions. Indeed, the paradox between off-shelf warming and on-shelf cooling in the Amundsen Sea is interesting. However, we still think that offshore pathways of CDW as well as the mechanisms, by which the mean ocean circulation warms the off-shelf CDW, are worth publishing. We did not intend to say that there is warming everywhere. In the revised manuscript, we show clearly that the warming and cooling is observed off and on the AS continental shelves, respectively (Fig. 3e and Lines 218-225).**

Furthermore, I think there are a few issues with the experimental design, or at least the presentation of that design, that need clarification. The authors conclude that the main determinant of the differences in the model simulations are the boundary conditions they apply. That effectively means that the explanation for what caused the changes lies not in the model results they are reporting, but in the larger-scale model from which the boundary conditions were taken. Very little text is devoted to those other results, and no explanation for the change in circulation is offered. The closest we get to an explanation comes on lines 143-147, but that only says that strong SAM "may" strengthen the Ross Gyre, and seems likely to increase upwelling. However, that seems to come from other work. Why not look at the ECCO results and tell the reader what they show?

*** In the previous version of this manuscript, we only showed the mean Ross Gyre strength between 2001-2006 and 2009-2014 in the supplementary material. In the revised manuscript, we include time series of Ross Gyre strength (based on ECCO LLC270 global optimization, Fig. S11). We also include an additional paragraph discussing the ECCO LLC270 global optimization (Lines 187-196).**

The other issue is the initial conditions, which are not mentioned at all in the main text. The methods section suggests that the initial conditions are the same for all experiments and are generated from a 10-year spin-up (with what boundary conditions?) of a model that starts from climatology. So what part of the 6-year runs reported in the paper is actually a transient response to a step change in boundary conditions? Are the CDW tracers continuously-released, or just introduced at the start? The figures suggest the latter. If that is the case, then the final distribution could be strongly dictated by the initial transient response, and that might be why the boundary conditions appear to exert the

dominant control over their final distribution.

*** We introduce the tracers at the start without restoring. Initial conditions are based on 10-year spin-up (2001-2010). In response to this comment, we run an additional two experiments (CTRL 2009-2014 and 2001-2006) with restoring tracers from the same region. As discussed in the supplementary sections (Lines 450-464 and Figs. S6 and S7), the spatial distributions of CDW tracers with restoring tracers are qualitatively similar to those experiments with initially-released tracers (Figs. 4 and S6). This indicates that the final distributions of the tracers, as well as off-shelf warming, is caused by boundary condition, not an initial transient response.**

Certainly Figure S6 suggests that the dominance of the boundary conditions over local surface forcing is not a robust or universal result. There are many times (in particular most of the time between year 1 and 4 on the time axis) when the difference between runs with different surface forcing is much larger than that between the runs with different boundary conditions.

*** We agree with the reviewer and have rephrased our conclusion to say that boundary conditions play an equally important role (Lines 23-25 and 140-154). We also clarify in the revised manuscript that off-shelf CDW warming develops over time with the warmest off-AS-shelf CDW during the last 2 years of simulation, similar to the tracer fields (Lines 125-128). Therefore, the impact of boundary condition becomes most apparent during the last 2 years.**

There are also problems with a number of the figures that really need to be addressed:

Figure 1: What projection is used to display the main panel? The edges of the model domain appear to be both curvilinear and asymmetric, and to extend beyond the displayed box. So the area covered by the model domain is never really clear. The inset suggests the edges follow parallels and meridians, so why do they look so distorted on the main panel? The same comment applies to all the map-plane figures that follow.

*** Figures 1, 3, 4, and 5 (in the main text) are displayed on a latitude-longitude grid showing the entire model domain. The LLC1080 grid uses a latitude-longitude grid north of 70°S and a bipolar grid south of 70°S (see Forget et al., 2015 for detailed description). Horizontal grid spacing in the AS and BS domain is 2-3 km. We add more information in the Methods section in the Supplementary material.**

Figure 3(c and d) and S3: It looks to me as if the inset boxes are incorrectly plotted on the main figure panels. That is misleading. The main panels indicate boxes that are on the continental shelf,

but temperatures and isobaths (?) in the inset suggest that the box might actually be somewhere to the north-east of where it is indicated to be.

*** We remove these boxes. In the revised manuscript, we include Fig. S10, which compares both off-shelf and on-shelf CDW properties.**

Figure S4: What is the source of these data? Why show so few stations from such a small region? If there really are only six stations, couldn't the comparison be shown in a more informative way? How about showing profiles? And what does the grey shading show? Presumably bathymetry, but what is the scale? Again, the form of the contours suggests that the box is not on the shelf.

***Thank you very much for pointing this out. In the revised manuscript, we use data from 2009 (Jacobs et al., 2010), 2010 (Nakayama et al., 2010), 2011 (Yager et al., 2012), 2012 (Dutrieux et al., 2012, Kim et al., 2017), and 2014 (Heywood et al., 2016). These citations are now included in the revised manuscript. We extend our data analysis and we analyze a number of CTD profiles in the eastern and central troughs (Lines 140-154 and Fig. S10). Figures with the grey scaling has been removed.**

Table S3 gives ranges in the last column, but didn't the cited paper give climatological mean values for each ice shelf? What do the ranges mean and where do they come from?

*** These ranges may be confusing, therefore we replaced these values with the ranges that are presented in Rignot et al., 2013.**

Reviewers' comments:

Reviewer #1 (Remarks to the Author):

My main criticisms of the original manuscript, mainly that the quantification of the impact of model boundary conditions on Amundsen Sea heat content is not established, remains unresolved. Short of this it is difficult to conclude that the results presented here are significant. Yes, boundary conditions matter, especially after long integration times, but that's not news. What would be novel would be a demonstration that this impacts melt, for example! Short of this, at least a demonstration that near ice ocean heat content is strongly affected is necessary. For reasons explained below, mainly involving figure S9, I don't think such a demonstration is made convincingly.

In addition I find the modified version of the manuscript to be difficult to read and full of inconsistencies/missing context/perhaps misunderstandings of some general concepts (see below). Therefore I cannot recommend this manuscript for publication at this point.

line 99-112: For clarity, the paragraph would benefit from a gentle rewrite. I found it very difficult to follow.

line 101: what is an 'isotracer'? Wouldn't it be easier to call it tracer concentration or define it as a measure of such?

line 102: remove 'clearly'?

line 102: does the sentence including 'clearly represent stronger westward transport' mean something like 'show that tracers originating from regions TR3 and TR4 reach their westernmost extent in CTRL(2009-2014).

Line 104: maybe introduce the idea that the tracer source has now changed? 'Notably, the origin of the CDW reaching the AS and BS also shifted westward in the 2001-2006 case.' then cite % numbers.

Line 109: ok, you conducted additional experiments and found that: it takes 5 years for ~50% of the tracers originating at the S04P section to reach the AS and BS? This is interesting but seems disconnected from the narrative, i.e. what is the point of this additional information?

Line 110; remove 'additional experiments' as this is a repeat from the line above?

Line 113: different boundary conditions vs different forcing: are you basically concluding that boundary conditions do matter, and they take ~5 years in your particular domain to do so, and that local (or domain) atmospheric forcing does not have a large effect on this influence? If so this is fine, though probably expected. But summarizing the idea in one sentence would help the narrative.

Also, this would make the story a bit more understandable. After all, the boundary conditions are set by past (slightly) remote and local forcing, so untangling local forcing and boundary conditions is difficult.

Line 123-...: Again, I find this hard to follow. Ok, so earlier you show that boundary conditions changes dictates changes in water sources for the AS and BS. Now you mean to quantify eventual impacts on AS and BS heat and freshwater contents, correct? If so you may want to clarify again and state this in introduction of the paragraph, following by a demonstration and your evidences.

Line 129: would it be fair, and perhaps clearer, to requalify 'the strengthening of the Ross Gyre in the domain' as the meridional expansion (or contraction) of the Ross Gyre in the domain?

Line 139: what is this warming due to? A change in depth of the isotherms? An actual change in the CDW characteristics (highly unlikely at these timescales)?

Line 140: Figure S9 is a really important figure, as it is the main quantification diagnostic of the impact of boundary conditions on AS heat content. It should be moved to the main text. But I disagree with the authors evaluation of the results. IF one chooses the period of the last 2 years to compare simulations, then maybe boundary conditions have a similar impact to that of local atmospheric forcing. In fact the best match with this assumption would be obtained by averaging over the period 3.5-4.5. But averaging over year 5 or most time interval would seemingly give much more ambiguous comparisons. In fact the graphic suggests that there are no significant differences between blue, magenta and cyan simulations in the eastern trough! Yet the magenta line has 'cold' boundary conditions for the reasons explained earlier by the authors. Surely that means that a major factor of heat variability at 409m depth there is of local atmospheric origin! That does not mean that boundary conditions do not matter. But it does suggest that it matters less than local forcing, not more.

What does figure 3e look like for the differences between the 4 experiments?

Also, I will note here that choosing one depth only to look at these processes is insufficient, as heaving of isotherms make it very difficult to relate signals to actual heat content changes.

Line 154: Here again looking at vertically integrated heat content would be a much better diagnostic to evaluate impacts near ice shelves. Presumably the anomaly propagates all the way there, otherwise what is the motivation for this exercise would disappear?

Line 155: why is the model evaluation section after the analysis??

Line 161: stronger mixing? Which kind of mixing? Do you have evidence for this?

The entire paragraph is highly speculative and do not describe discrepancies between model and observations adequately. One major miss is the representation of the upper water column AND thermocline depth, which is much shallower in the model than in the observations. This is typical for models, but cannot NOT be mentioned, especially since this is likely to have major impacts on the conclusions!

Line 180: change title from 'Discussion' to, say, 'Boundary conditions variability'

Line 185: what is the ECCO LLC270, references, description, point to supplementary material?

Line 188: how is the Ross gyre strength defined?

Line 198: It may be the case that forcing outside of the model domain play a role, but it could also be the case that past forcing inside the model domain led to the boundary conditions... Likely, it's a mix of the two.

Line 200: the connection with ENSO or SAM doesn't make sense here: why do we need to invoke a larger domain to talk about ENSO or SAM? Can't they have an impact locally as well/in addition?

Line 208: ENSO and SAM are not independent indices...

Line 215: What is large scale atmosphere? Do atmospheric wave trains participate in this large scale atmosphere? Were previous studies talking about small scale atmosphere? Or do you mean that the scale of the atmospheric anomaly is similar, but that you are looking at large scale ocean response, instead of localized ocean response?

Line 218: a very surprising results! As mentioned earlier, I don't think you have actually demonstrated that it is indeed true. Assuming it is, however, it seems to contradict the few

observational evidences we have... But the model doesn't need to be perfect to be useful. Could it be that in the real world (and it this model as well, but see earlier comments on this) boundary conditions play a minor role in modulating the strength of the AS/southern ocean exchanges, and ultimately the heat content in the AS?

Line 231: CDW properties are NOT set by the Ross gyre strength. Please revise wording, probably by simply talking about modified CDW, or the sources of the modified CDW on the continental shelf. I will note here again that a 0.2C change at one depth level is easily obtained by a change in thermocline depth, and that you have not demonstrated that this is not what is happening here. If that were the case this change may have minimal or very indirect impact on AS and BS heat content.

Reviewer #2 (Remarks to the Author):

Overall, the manuscript is improved relative to the initial submission. The primary finding is that the source of CDW flowing onto the continental shelves in the Amundsen and Bellingshausen seas in the model is from a region to the north, and that this source varies depending on the far-field oceanographic conditions that constitute the lateral boundary conditions for the model, rather than local atmospheric conditions. Over long timescales in particular, this may influence the melt rate of ice shelves in the Amundsen and Bellingshausen seas, and hence sea-level rise, so is of relevance to the wider audience. The findings regarding pathways of CDW towards the continental shelf and the controls on the properties of this CDW may influence future studies in this field. In my opinion, these findings are worthy of publication, but the manuscript needs modification to address the following comments:

1. I feel that the introduction should mention the along slope currents, as described in e.g., Walker et al., 2013, and the ability of the model to replicate this current and the onshore flux of CDW through key troughs.
2. I would still like to know whether the total amount of CDW fluxed onto the shelf is different between the CTRL and 2001-2006 simulations – at present, only the change in CDW property and source are discussed.
3. The model thermocline is clearly too shallow, by some distance. How does this affect the pathway of CDW onto the continental shelf, and the conclusions presented about the origin of this water? Also, this would seem likely to impact the melt rate of the glaciers, but this is not discussed at all.
4. Lines 75-77: I assume you have done some spatial averaging or integration of the tracers to determine the fractional contribution of each. However, the boundaries of the Amundsen Sea used for this calculation should be defined, as it may be that the exact contribution of each tracer is domain-dependent.
5. Figure 4 and similar: the colorbar/legend really needs to be placed outside of the figure panels and larger, it is not easily legible at present.
6. It is not clear (to me) why in the restoring tracers experiments, a value of 1.0 is used as opposed to 100, and also why the resultant integrated tracer figures (S4, S6, S7) have colorbars ranging from 1e-3 to 1.0, compared with 1e2 to 1e5, different by a factor of 1e5, and not simply 100. If anything, I would have expected more total tracer in the experiment with restored tracers than the initial release experiment. What am I missing?
7. Line 143 (and 146): "Different forcing for the last 2 years of model simulations"? I think you mean, the temperature difference over the last 2 years of the model simulations between the CTRL

and the BC09AF01 simulations, but I initially read this statement and thought the atmospheric forcing had only been changed for the last two years. Please rephrase

8. Line 160-163: This statement seems to contradict both Dutrieux et al. (2014) and Webber et al. (2017). I think the discrepancy is that here the authors analyse the temperature of the deep CDW, while those papers primarily investigate the depth of the thermocline and the volume flux of CDW. Which is more important is not clear, but there should be some discussion of the depth of the thermocline between 2009-2011 and 2012-2014.

Minor comments:

Line 26: replace southern with southward

Line 113: Qualify this statement with "main/principle/primary" (or similar) before "reason"

Line 185: Replace "primary" with "primarily"

Lines 455-458: Specify that the tracers are restored in the regions indicated by purple boxes in Figs XX.

Response to the specific comments

(Comments from reviewer are in italics; our responses are indicated in bold typeface)

Reviewers' comments:

Reviewer #1 (Remarks to the Author):

My main criticisms of the original manuscript, mainly that the quantification of the impact of model boundary conditions on Amundsen Sea heat content is not established, remains unresolved. Short of this it is difficult to conclude that the results presented here are significant. Yes, boundary conditions matter, especially after long integration times, but that's not news. What would be novel would be a demonstration that this impacts melt, for example! Short of this, at least a demonstration that near ice ocean heat content is strongly affected is necessary. For reasons explained below, mainly involving figure S9, I don't think such a demonstration is made convincingly.

In addition I find the modified version of the manuscript to be difficult to read and full of inconsistencies/missing context/perhaps misunderstandings of some general concepts (see below). Therefore I cannot recommend this manuscript for publication at this point.

We thank the reviewer for the careful review of our manuscript and helpful and insightful comments. We agree that our explanation was not clear. To reduce confusion and more convincingly emphasize the key points of this manuscript, we simplified old Fig. 3, removed old Fig. S9 and include new Fig. 7, which shows time series of spatially averaged on-shelf CDW temperature for the eastern AS region near Pine Island Glacier. Fig. 6 clearly shows the impact of boundary forcing on warming off-shelf CDW waters. Fig. 7 demonstrates that this off-shelf warming impacts on-shelf CDW temperature.

In our opinion, the prediction that far-field ocean properties and circulation (i.e., the boundary condition in our regional simulation) matter for CDW conditions off-shelf from the AS and BS is novel and significant. We show that CDW intruding onto the AS and BS continental shelves in the numerical simulations comes from a region to the north and that this source property varies depending on oceanographic conditions. To our knowledge, no previous study has shown how source CDW properties are determined. Based on available observations and numerical simulations, this study shows, for the first time, that CDW flowing onto the AS and BS continental shelf regions is strongly influenced by large-scale ocean circulation.

line 99-112: For clarity, the paragraph would benefit from a gentle rewrite. I found it very difficult to follow.

We modified this paragraph (Lines 100-113) as per comments below.

line 101: what is an 'isotracer'? Wouldn't it be easier to call it tracer concentration or define it as a measure of such?

We replace "isotracer contour" with "contours of vertically integrated tracer concentration".

line 102: remove 'clearly'?

We revised the manuscript as suggested.

line 102: does the sentence including 'clearly represent stronger westward transport' mean something like 'show that tracers originating from regions TR3 and TR4 reach their westernmost extent in CTRL(2009-2014).

We removed this sentence in the revised manuscript.

Line 104: maybe introduce the idea that the tracer source has now changed? 'Notably, the origin of the CDW reaching the AS and BS also shifted westward in the 2001-2006 case.' then cite % numbers.

We followed reviewer's advice in revised manuscript.

Line 109: ok, you conducted additional experiments and found that: it takes 5 years for ~50% of the tracers originating at the S04P section to reach the AS and BS? This is interesting but seems disconnected from the narrative, i.e. what is the point of this additional information?

In the revised manuscript, we explain that CDW tracer pathways remain similar throughout the model integration for other sensitivity experiments (Lines 110-113).

Line 110; remove 'additional experiments' as this is a repeat from the line above?

We removed the first "additional".

Line 113: different boundary conditions vs different forcing: are you basically concluding that boundary conditions do matter, and they take ~5 years in your particular domain to do so, and that local (or domain) atmospheric forcing does not have a large effect on this influence? If so this is fine, though probably expected. But summarizing the idea in one sentence would help the narrative. Also, this would make the story a bit more understandable. After all, the boundary conditions are set

by past (slightly) remote and local forcing, so untangling local forcing and boundary conditions is difficult.

We revised the manuscript as suggested (Lines 114-116).

Line 123-...: Again, I find this hard to follow. Ok, so earlier you show that boundary conditions changes dictate changes in water sources for the AS and BS. Now you mean to quantify eventual impacts on AS and BS heat and freshwater contents, correct? If so you may want to clarify again and state this in introduction of the paragraph, following by a demonstration and your evidences.

For more clarity, we added a section (Reason for off-shelf CDW warming) and modified Lines 126-136.

Line 129: would it be fair, and perhaps clearer, to requalify 'the strengthening of the Ross Gyre in the domain' as the meridional expansion (or contraction) of the Ross Gyre in the domain?

We think it is appropriate to use "the strength of the Ross Gyre". Based on Fig. S9, we do not see obvious structural changes. However, Ross Gyre strengths are 28 Sv and 35 Sv for 2001-2006 and CTRL(2009-2014) cases, respectively.

Line 139: what is this warming due to? A change in depth of the isotherms? An actual change in the CDW characteristics (highly unlikely at these timescales)?

Warming shown here is at 409 m, which is well below thermocline depth (Fig. S1). This warming at the northern model boundary is likely caused by slight southward shift of ACC (Fig. S9). We revise the manuscript as in Lines 147-148.

Line 140: Figure S9 is a really important figure, as it is the main quantification diagnostic of the impact of boundary conditions on AS heat content. It should be moved to the main text. But I disagree with the authors evaluation of the results. IF one chooses the period of the last 2 years to compare simulations, then maybe boundary conditions have a similar impact to that of local atmospheric forcing. In fact the best match with this assumption would be obtained by averaging over the period 3.5-4.5. But averaging over year 5 or most time interval would seemingly give much more ambiguous comparisons. In fact the graphic suggests that there are no significant differences between blue, magenta and cyan simulations in the eastern trough! Yet the magenta line has 'cold' boundary conditions for the reasons explained earlier by the authors. Surely that means that a major factor of heat variability at 409m depth there is of local atmospheric origin! That does not mean that boundary conditions do not matter. But it does suggest that it matters less than local forcing, not more.

We replace old Fig. S9 with new Fig. 7. This figure clearly shows that on-shelf CDW temperature is controlled by lateral boundary condition and thus far field ocean circulation and properties (Lines 150-165).

What does figure 3e look like for the differences between the 4 experiments?

We add these figures in the revised manuscript (Fig. 6).

Also, I will note here that choosing one depth only to look at these processes is insufficient, as heaving of isotherms make it very difficult to relate signals to actual heat content changes.

We revised the manuscript as suggested (Fig. 7).

Line 154: Here again looking at vertically integrated heat content would be a much better diagnostic to evaluate impacts near ice shelves. Presumably the anomaly propagates all the way there, otherwise what is the motivation for this exercise would disappear?

We included a discussion related to vertically integrated heat content for the AS and BS domains (Lines 166-178, Fig. S10). However, the vertically integrated heat content is not a very good diagnostic for evaluating the impact on on-shelf CDW properties because integrated heat content is strongly controlled by its volume, i.e., thermocline depth. In order to see the impact on CDW properties, we instead analyze domain averaged CDW temperature.

Line 155: why is the model evaluation section after the analysis??

We revised the manuscript as suggested.

Line 161: stronger mixing? Which kind of mixing? Do you have evidence for this?

Nakayama et al. 2013 show that on-shelf CDW shows large variabilities horizontally and vertically (e.g., Figs., 8 and 11 in Nakayama et al., 2013). We remove “stronger mixing” and now state “on-shelf CDW properties are highly variable horizontally and vertically”.

The entire paragraph is highly speculative and do not describe discrepancies between model and observations adequately. One major miss is the representation of the upper water column AND thermocline depth, which is much shallower in the model than in the observations. This is typical for models, but cannot NOT be mentioned, especially since this is likely to have major impacts on the conclusions!

We include a discussion on simulated too thin WW layer (Lines 202-206). In this paper, important conclusion is that “far field ocean properties and circulation dominantly control off-shelf CDW properties, which intrudes onto the AS and BS continental shelves”. This fact

is supported by the fact that that data and model both show warmer off- and on-shelf CDW properties between 2012-2014.

Line 180: change title from 'Discussion' to, say, 'Boundary conditions variability'

We revised the manuscript as suggested.

Line 185: what is the ECCO LLC270, references, description, point to supplementary material?

Thank you for pointing it out. We added the description and reference for ECCO LLC270 in supplementary material.

Line 188: how is the Ross gyre strength defined?

We now include this information in the main text (Line 221).

Line 198: It may be the case that forcing outside of the model domain play a role, but it could also be the case that past forcing inside the model domain led to the boundary conditions... Likely, it's a mix of the two.

It is correct that past forcing inside the model domain can lead to the changes we see at the model lateral boundary. We revised the manuscript (Lines 231-233).

Line 200: the connection with ENSO or SAM doesn't make sense here: why do we need to invoke a larger domain to talk about ENSO or SAM? Can't they have an impact locally as well/in addition?

In this manuscript, we show that off-shelf CDW warming is primarily controlled by large-scale ocean circulation and changes in mid-latitude CDW properties. Since El Niño-Southern Oscillation (ENSO) and Southern Annular Mode (SAM) variabilities have large impact on the atmospheric circulation over the Southern Ocean, we include this discussion related to SAM and ENSO. We have revised the manuscript (Lines 233-236).

Line 208: ENSO and SAM are not independent indices...

Replaced "and" with "or" in "SAM or ENSO" for clarity.

Line 215: What is large scale atmosphere? Do atmospheric wave trains participate in this large scale atmosphere? Were previous studies talking about small scale atmosphere? Or do you mean that the scale of the atmospheric anomaly is similar, but that you are looking at large scale ocean response, instead of localized ocean response?

We mean the latter. We revised the manuscript as suggested (Lines 249-252).

Line 218: a very surprising results! As mentioned earlier, I don't think you have actually demonstrated that it is indeed true. Assuming it is, however, it seems to contradict the few observational evidences we have... But the model doesn't need to be perfect to be useful. Could it be that in the real world (and it this model as well, but see earlier comments on this) boundary conditions play a minor role in modulating the strength of the AS/southern ocean exchanges, and ultimately the heat content in the AS?

We include new Fig. 7 showing that on-shelf CDW temperature is controlled mainly by lateral boundary condition.

Line 231: CDW properties are NOT set by the Ross gyre strength. Please revise wording, probably by simply talking about modified CDW, or the sources of the modified CDW on the continental shelf. I will note here again that a 0.2C change at one depth level is easily obtained by a change in thermocline depth, and that you have not demonstrated that this is not what is happening here. If that were the case this change may have minimal or very indirect impact on AS and BS heat content.

We revised the manuscript as suggested. We also include new Fig. 7.

Reviewer #2 (Remarks to the Author):

Overall, the manuscript is improved relative to the initial submission. The primary finding is that the source of CDW flowing onto the continental shelves in the Amundsen and Bellingshausen seas in the model is from a region to the north, and that this source varies depending on the far-field oceanographic conditions that constitute the lateral boundary conditions for the model, rather than local atmospheric conditions. Over long timescales in particular, this may influence the melt rate of ice shelves in the Amundsen and Bellingshausen seas, and hence sea-level rise, so is of relevance to the wider audience. The findings regarding pathways of CDW towards the continental shelf and the controls on the properties of this CDW may influence future studies in this field. In my opinion, these findings are worthy of publication, but the manuscript needs modification to address the following comments:

Thank you very much for encouraging and insightful comments.

1. I feel that the introduction should mention the along slope currents, as described in e.g., Walker et al., 2013, and the ability of the model to replicate this current and the onshore flux of CDW through key troughs.

We revised the manuscript as suggested (Lines 46-47).

2. *I would still like to know whether the total amount of CDW fluxed onto the shelf is different between the CTRL and 2001-2006 simulations – at present, only the change in CDW property and source are discussed.*

In the revised manuscript we did not include the analysis on the total amount of CDW fluxed on the shelf but we include (1) time series of averaged CDW temperature (Lines 150-165, Fig.7) and (2) a discussion on vertically integrated on-shelf heat content for AS and BS continental shelves (Lines 166-178, Fig.S10).

3. *The model thermocline is clearly too shallow, by some distance. How does this affect the pathway of CDW onto the continental shelf, and the conclusions presented about the origin of this water? Also, this would seem likely to impact the melt rate of the glaciers, but this is not discussed at all.*

We include additional discussion about too shallow thermocline depth near the continental shelf break (Lines 202-206). We do not include discussion on the impact on the ice shelf melt rates because we think our simulations are not long enough to see the impact. We think that ice shelf melt rates are controlled by many different processes, and longer model integration time is likely required to see the impact from off-shelf CDW temperature change.

4. *Lines 75-77: I assume you have done some spatial averaging or integration of the tracers to determine the fractional contribution of each. However, the boundaries of the Amundsen Sea used for this calculation should be defined, as it may be that the exact contribution of each tracer is domain-dependent.*

We modified Fig. 1 and include the regions used to integrate the tracer concentration to calculate the fractional contribution. We have slightly modified the boundaries of the AS and BS.

5. *Figure 4 and similar: the colorbar/legend really needs to be placed outside of the figure panels and larger, it is not easily legible at present.*

We revised the manuscript as suggested.

6. *It is not clear (to me) why in the restoring tracers experiments, a value of 1.0 is used as opposed to 100, and also why the resultant integrated tracer figures (S4, S6, S7) have colorbars ranging from 1e-3 to 1.0, compared with 1e2 to 1e5, different by a factor of 1e5, and not simply 100. If anything, I would have expected more total tracer in the experiment with restored tracers than the initial release experiment. What am I missing?*

Since we think it is more common to restore tracer concentration to 1.0, we use a value of 1.0 as opposed to 100. No special reasons. Figs. 4 and S4 shows vertically integrated tracer concentration. Since initial tracer concentration is 100 and CDW layer is 500-1000 m (Fig. 2), color scales are $\sim 10^5$ times larger in Figs., 4 and S4 than those in Figs. S5 and S6.

7. Line 143 (and 146): “Different forcing for the last 2 years of model simulations”? I think you mean, the temperature difference over the last 2 years of the model simulations between the CTRL and the BC09AF01 simulations, but I initially read this statement and thought the atmospheric forcing had only been changed for the last two years. Please rephrase.

We revised the manuscript as suggested (Line 157-158).

8. Line 160-163: This statement seems to contradict both Dutrieux et al. (2014) and Webber et al. (2017). I think the discrepancy is that here the authors analyze the temperature of the deep CDW, while those papers primarily investigate the depth of the thermocline and the volume flux of CDW. Which is more important is not clear, but there should be some discussion of the depth of the thermocline between 2009-2011 and 2012-2014.

We revised the manuscript as suggested (Line 166-178).

Minor comments:

Line 26: replace southern with southward

We revised the manuscript as suggested.

Line 113: Qualify this statement with “main/principle/primary” (or similar) before “reason”

As suggested, we replaced “The reason” with “The main reason”.

Line 185: Replace “primary” with “primarily”

We revised the manuscript as suggested.

Lines 455-458: Specify that the tracers are restored in the regions indicated by purple boxes in Figs XX.

We revised the manuscript as suggested.

REVIEWERS' COMMENTS:

Reviewer #1 (Remarks to the Author):

The manuscript improved a lot. It is now clear, and pending corrections of a few typos is acceptable for publication.

Here are some suggestions:

line 16: from 'the' Antarctic Continent

line 22: in 'the' 2009-104 case than in 'the' 2001-2006 case

line 26: model domain 'are' able to control

line 41: be measured again in 2018 -> As this been done over the last few months? or is this planned for next season? In the former case, perhaps reword.

line 44: but with 'a' nominal horizontal

line 60: melt rates ', ' in good agreement with

line 74: southwestward seems contradictory to the southeastward above, perhaps simply replace by southwestward by 'southward'.

line 75: 'isopycnals'

line 85: southward 'in' the region

line 87: remove 'there exists' and 'that'

line 89: from the 'weaker' circulation

line 90: (fig. 3a) 'in' the CTRL(2009-2014) case

line 102: continental shelves 'originates' from further west

line 108-109: above you were comparing 2009-2014 to 2001-2006. Not here? Is there a difference in the BS between this case and CTRL?

line 110: remove ', for the 2001-2006 case'

line 112: remove 'other'

line 134: remove 'off the AS'

line 186: is ref 19 the most suited? Presumably there are a few others that actually concentrate on continental shelf edge spatial and temporal variability?

line 201: We note that simulated WW 'in these regions is thinner than observations by 100-200 m'

line 237: time-mean SAM 'index is' 0.21, ...

line 512: a concentration 'of' 1.0 at every

Reviewer #2 (Remarks to the Author):

Overall, the authors have responded to all my scientific concerns. However, there are some issues with the language used in the revised document, so I suggest minor presentational corrections. I have focussed on the added text, but more thorough editing may be necessary.

Line 12: CDW is not the only cause of melting. Suggest change "caused" to "enhanced" or "increased". Or reword the sentence (e.g., CDW leads to melting of WA ice shelves).

Line 102: change "is originated" to "originates"

Line 128: Start sentence with "The"

Line 130: "those in" confused me. Just delete (i.e., change to "different from the ...")

Line 133: change "behaviors" to "the behavior"

Line 152-164: This paragraph is too dense and filled with sentences that are too long. Please re-

write, with each sentence introducing one concept – for example, Line 152-157 includes two sub-definitions and nested brackets. Separate this into separate sentences and build up a logical narrative. Also, I don't like the use of "by calculating the differences..." at the end of an already long sentence. Perhaps "differences are... 0.04 °C (CTRL(2009-2014) – BC09AF01) and 0.004 °C (BC01AF09 – 2001-2006)" is better? I'm not sure. Alternatively, state the calculation made first, then the results. A table might help?

Line 168: "reduction of *the* melt rate"

Line 174-177: This paragraph is necessary, but the last two sentences don't really make sense in the context of the preceding calculations. I think the point you need to argue is that it may not be the vertically-integrated heat content that really matters, and instead that the temperature at the grounding line may be more important to glacial retreat. I'm not sure that the question of which is more important is yet resolved in the literature, but references would help support your argument.

Response to the specific comments

(Comments from reviewer are in italics; our responses are indicated in bold typeface)

Reviewers' comments:

Reviewer #1 (Remarks to the Author):

The manuscript improved a lot. It is now clear, and pending corrections of a few typos is acceptable for publication.

Here are some suggestions:

line 16: from 'the' Antarctic Continent

We revised the manuscript as suggested.

line 22: in 'the' 2009-104 case than in 'the' 2001-2006 case

We revised the manuscript as suggested.

line 26: model domain 'are' able to control

We revised the manuscript as suggested.

line 41: be measured again in 2018 -> As this been done over the last few months? or is this planned for next season? In the former case, perhaps reword.

We revised the manuscript as suggested.

line 44: but with 'a' nominal horizontal

We revised the manuscript as suggested.

line 60: melt rates ',' in good agreement with

We revised the manuscript as suggested.

line 74: southwestward seems contradictory to the southeastward above, perhaps simply replace by southwestward by 'southward'.

We revised the manuscript as suggested.

line 75: 'isopycnals'

We revised the manuscript as suggested.

line 85: southward 'in' the region

We revised the manuscript as suggested.

line 87: remove 'there exists' and 'that'

We revised the manuscript as suggested.

line 89: from the 'weaker' circulation

We revised the manuscript as suggested.

line 90: (fig. 3a) 'in' the CTRL(2009-2014) case

We revised the manuscript as suggested.

line 102: continental shelves 'originates' from further west

We revised the manuscript as suggested.

line 108-109: above you were comparing 2009-2014 to 2001-2006. Not here? Is there a difference in the BS between this case and CTRL?

We revised the manuscript as suggested.

line 110: remove ', for the 2001-2006 case'

We revised the manuscript as suggested.

line 112: remove 'other'

We revised the manuscript as suggested.

line 134: remove 'off the AS'

We revised the manuscript as suggested.

line 186: is ref 19 the most suited? Presumably there are a few others that actually concentrate on continental shelf edge spatial and temporal variability?

Nakayama et al., 2013 argue that on-shelf CDW properties are highly variable horizontally and vertically (see Fig. 11 and section 4.2). In the revised manuscript, we also include Assmann et al., 2013, because they argue the variability of CDW transport using mooring records at the similar location.

line 201: We note that simulated WW 'in these regions is thinner than observations by 100-200 m'
We revised the manuscript as suggested.

line 237: time-mean SAM 'index is' 0.21, ...
We revised the manuscript as suggested.

line 512: a concentration 'of' 1.0 at every
We revised the manuscript as suggested.

Reviewer #2 (Remarks to the Author):

Overall, the authors have responded to all my scientific concerns. However, there are some issues with the language used in the revised document, so I suggest minor presentational corrections. I have focussed on the added text, but more thorough editing may be necessary.

Line 12: CDW is not the only cause of melting. Suggest change “caused” to “enhanced” or “increased”. Or reword the sentence (e.g., CDW leads to melting of WA ice shelves).
We revised the manuscript as suggested.

Line 102: change “is originated” to “originates”
We revised the manuscript as suggested.

Line 128: Start sentence with “The”
We revised the manuscript as suggested.

Line 130: “those in” confused me. Just delete (i.e., change to “different from the ...”)
We revised the manuscript as suggested.

Line 133: change “behaviors” to “the behavior”
We revised the manuscript as suggested.

Line 152-164: This paragraph is too dense and filled with sentences that are too long. Please re-write, with each sentence introducing one concept – for example, Line 152-157 includes two sub-definitions and nested brackets. Separate this into separate sentences and build up a logical

narrative. Also, I don't like the use of "by calculating the differences..." at the end of an already long sentence. Perhaps "differences are... 0.04 °C (CTRL(2009-2014) – BC09AF01) and 0.004 °C (BC01AF09 – 2001-2006)" is better? I'm not sure. Alternatively, state the calculation made first, then the results. A table might help?

For lines 153-158, the original sentence is separated into two sentences. For lines 158-166, we modified the manuscript as suggested.

*Line 168: "reduction of *the* melt rate"*

We revised the manuscript as suggested.

Line 174-177: This paragraph is necessary, but the last two sentences don't really make sense in the context of the preceding calculations. I think the point you need to argue is that the it may not be the vertically-integrated heat content that really matters, and instead that the temperature at the grounding line may be more important to glacial retreat. I'm not sure that the question of which is more important is yet resolved in the literature, but references would help support your argument.

We revised the manuscript as suggested (Lines 177-179).